# Heterogeneous Skill Learning for Multi-agent Tasks

**Yuntao Liu**
Academy of Military Science
Beijing, China
liu-yt@foxmail.com

**Yuan Li**
Academy of Military Science
Beijing, China
liyuan@nudt.edu.cn

**Xinhai Xu** *
Academy of Military Science
Beijing, China
xuxinhai@nudt.edu.cn

**Yong Dou**
National University of Defense Technology
Hunan Changsha, China
douyong@nudt.edu.cn

**Donghong Liu** *
Academy of Military Science
Beijing, China
liu_donghong@sina.com

## Abstract

Heterogeneous behaviours are widespread in many multi-agent tasks, which have not been paid much attention in the community of multi-agent reinforcement learning. It would be a key factor for improving the learning performance to efficiently characterize and automatically find heterogeneous behaviours. In this paper, we introduce the concept of the skill to explore the ability of heterogeneous behaviours. We propose a novel skill-based multi-agent reinforcement learning framework to enable agents to master diverse skills. Specifically, our framework consists of the skill representation mechanism, the skill selector and the skill-based policy learning mechanism. We design an auto-encoder model to generate the latent variable as the skill representation by incorporating the environment information, which ensures the distinguishable of agents for skill selection and the discriminability for skill learning. With the representation, a skill selection mechanism is invented to realize the assignment from agents to skills. Meanwhile, diverse skill-based policies are generated through a novel skill-based policy learning method. To promote efficient skill discovery, a mutual information based intrinsic reward function is constructed. Empirical results show that our framework obtains the best performance on three challenging benchmarks, i.e., StarCraft II micromanagement tasks, Google Research Football and GoBigger, over state-of-the-art MARL methods.

## 1 Introduction

The cooperative multi-agent reinforcement learning (MARL) has obtained great improvements in many multi-agent systems, such as real-time strategy games [2], artificial swarms of robots [13] and autonomous vehicles [14]. The original solution is the independent learning method, in which each agent treats other agents as part of the environment. The main drawback is that simultaneous learning of all agents will result in a non-stationary environment from the perspective of each agent [3]. To avoid this problem, fully centralized learning approaches [18] have been studied, which aim to learn an optimal joint action of agents in response to the joint observation. However, the solution space grows exponentially with the number of agents, making it hard to find a good policy in a limited time [4]. Later, following the *centralized training with decentralized execution* (CTDE) paradigm, value decomposition based MARL methods have been proposed which attracts wide attention [21, 26, 33, 31]. The main idea is that each agent learns its policy based on local observation while a centralized mechanism is designed to guide the training of all agents. In most such methods,

---

*Corresponding authors.

36th Conference on Neural Information Processing Systems (NeurIPS 2022).

agents share their neural networks to compute actions, which is known as the parameter sharing technique. In this way, the training efficiency is significantly improved due to the reduction of the number of parameters, and enough samples shared by all agents in the centralized training procedure. However, heterogeneous behaviours are hard to be generated for such kind of methods when facing complex multi-agent tasks.

To address this challenge, some recent methods learn different roles for agents based on their observations and actions. Agents are grouped into different roles which carry out different tasks. ROMA [27] assumes a Gaussian distribution for learning roles and RODE [28] uses a pre-trained model with random samples to restrict the action space for each role. However, roles are sometimes not easy to be distinguished based on only observation-action pairs. Moreover, the diversity of heterogeneous behaviours is hard to be fully explored. Recently, CDS [4] introduces diversity into the MARL approach by adding an extra local network for each agent to the shared network. It tries to make a balance between diversity and the high learning efficiency brought by parameter sharing. However, extra introduced networks for all agents hinder the application of CDS on large-scale tasks, which will be illustrated in the experiment.

Different from previous works, we consider studying heterogeneous behaviours in multi-agent tasks by introducing the skill learning [20]. The main idea is to enable agents to learn diverse skills, and then select proper skills for agents to complete certain tasks. For instance, in Google Research Football game, if multiple useful skills such as off-the-ball moving, dribbling with the ball, passing and shooting can be learned, agents can later use these skills to easily cooperate to win the match in a short policy search time. Introducing skill learning into MARL approaches has two advantages. On one hand, distinct skill representation will help to generate diverse policies and enable agents to select heterogeneous behaviours. On the other hand, the skill learning process is compatible with the parameter sharing technique, which does not bring much overhead for training efficiency. Then three questions arise: (1) how skills are represented; (2) how skills are learned;(3) how agents select proper skills.

There are some works studying the problem of skill generation in the single-agent area. Early works used the human-designed reward function to discover skills [11]. However, expert knowledge is often difficult to acquire. Autonomous acquisition of useful skills without human-supervised signals can be divided into two types. One is adopting mutual information maximization for skill discovery [6]. It uses model-free unsupervised RL methods to learn distinguishable skills which can perform distinct tasks. Other works [5, 19] consider skill discovery with intrinsic motivation which drives the exploration of agents. However, for those works, skills are learned one by one from different single-agent scenarios. This is quite different from multi-agent tasks, in which different kinds of skills must be learned in one scenario. Moreover, multiple agents should make decisions on the selection of skills and the decision will change as the game goes on. For multi-agent tasks, HSD [32] studies the problem of skill discovery with a hierarchical RL method. However, it does not make full consideration of distinguishable and diverse skill discovery, leading to poor performance for heterogeneous problems.

To learn effective skills in multi-agent tasks, we propose a novel skill-based learning framework. We design an auto-encoder model to generate the latent variable as the skill representation. To reflect the potential characters of agents and ensure diversity of skills, a set of skill-ids as well as observations and actions of agents are used as the input while the reward of the environment and the next observations are used as the output. With this representation, we devise a skill selection mechanism that assigns an agent with a skill. Due to characters encoded in the skill representation, this mechanism could select proper skills for agents based on their observations. Finally, we introduce the information-theoretic paradigm of mutual information to discover distinguishable and diverse skills. The optimization objective of mutual information is further represented as an intrinsic reward, which together with the environment reward is used to perform skill-based policy learning. For skill-based policy learning, we still adopt the parameter sharing technique, but we further compute the latent variable conditioned policy for each skill. In this way, our framework learns distinguishable skills and encourages each skill to explore the environment and perform its task more effectively.

We make extensive experiments to evaluate our framework on three challenging benchmarks, i.e., StarCraft II micromanagement environment, Google Research Football and GoBigger. To our best knowledge, the proposed framework achieves state-of-the-art performance in all tasks.

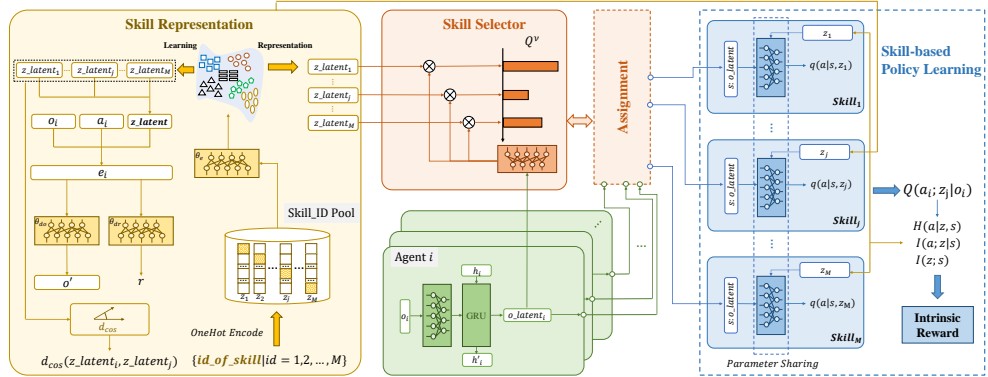

Figure 1: Schematics of HSL. It consists of three parts: skill representation, skill selector and skill-based policy learning. The skill representation generates latent variables $z\_latent$ for the skill selector. The skill selector is equivalent to an assignment module, which helps to select skills for agents. The last part is responsible to generate diverse skills.

## 2  Background

**Multi-agent Markov Games**: We consider a partially observable multi-agent game, which can be modeled as a multi-agent extension of Markov Decision Processes [34]. It is denoted by $\langle N, S, O, \{A_i\}_{i=1}^n, \{R_i\}_{i=1}^n, P, \gamma \rangle$, where $\mathcal{N}$ is a finite set of $n$ agents. $S$ is the state space, $O$ is the partial observation space of agents, $A_i$ is the action of agent $i$, $R_i : S \times A_1 \times \cdots \times A_N \to \mathfrak{R}$ is the reward function of agent $i$ and $P = S \times A_1 \times \cdots \times A_N \times S \to [0, 1]$ is the state transition function. A popular solution approach for multi-agent games is the value decomposition based MARL. Each agent $i$ is associated with a neural network, which is used to compute the Q-value $Q_i$ based on its local observation $o_i$. During the training process, a global Q-value $Q_{tot}$ is computed as a function of $Q_i$, for example, the summation used in VDN [24] and the mixing network used in QMIX [21].

**Mutual information**: Mutual information is a core quantity in the information-theory that measures the dependence between random variables using a Shannon entropy-based approach, defined as Equation (1).

$$\mathcal{I}(X; Y) = \mathcal{H}(X) - \mathcal{H}(X|Y) \tag{1}$$

According to the definition, maximizing the mutual information $\mathcal{I}(\cdot; \cdot)$ means maximizing the entropy $\mathcal{H}(X)$ and minimizing the conditional entropy $\mathcal{H}(X|Y)$ at the same time.

In the context of reinforcement learning, $X$ refers to the function of the state and $Y$ is the function of actions. Maximizing the mutual information encourages the entropy of the state to be high, encouraging exploration in policy learning. Skill-based RL methods such as [6, 5] adopt the mutual information to perform diverse skill discovery.

## 3  Method

In this section, we present a novel heterogeneous skill learning MARL framework (HSL) method for multi-agent tasks. The framework is shown in Figure 1. We introduce the concept skill which is a latent-conditioned ($Z$ conditioned) policy $\pi(A|S, Z)$ to enable agents to perform diverse behaviours. Suppose there are $M$ skills, of which ids are encoded in a one-hot manner, and they are denoted by $z_j, j = 1, 2, ...M$. We first encode the environment information, i.e, observations and actions of agents, with skill-ids as latent skill variables $z\_latent_j, j = 1, 2, ...M$. This process is called skill representation. Based on latent variables as key features, we design a skill assignment mechanism, which associates each agent with a skill such that agents could learn heterogeneous policies. With skill representations and skill assignments, diverse skill policies are learned through the skill learning mechanism.

## 3.1 Skill Representation Learning

Skill representation learning generates latent skill variables, which are used for skill assignment and skill learning. On one hand, generated latent skill variables should reflect the characters of different agents, which helps agents make proper choices over different skills. On the other hand, they provide key information that makes the learned skills as different as possible.

To achieve the aforementioned goals, we utilize key information from the environment such as the environmental state transition function and the reward function for skill representation. We design an auto-encoder model to generate skill variables. The encoder is denoted by $f_e(\cdot; \theta_e)$ where $\theta_e$ represents parameters of the model. It encodes the one-hot vector of skill-id $z_j$ to the latent skill variable $z\_latent_j = f_e(z_j; \theta_e)$. There are two decoders with the same input which is a concatenation of the observation $o_i$, the action $a_i$ of agent $i$ and all latent skill variables $z\_latent = \{z\_latent_j | j = 1, 2, ..., M\}$. The output of one decoder, denoted by $f_{dr}(\cdot; \theta_{dr})$, is the estimation of environment reward while that of the other decoder, denoted by $f_{do}(\cdot; \theta_{do})$, is the estimation of next observation of agent $i$. The learning objective of the auto-encoder model is shown as Equation (2). With the first two terms, the auto-encoder model could generate latent skill variables which reveal the effect of different skills on observations, actions and the environment. Consequently, it is beneficial for the skill selector to distinguish agents when their states and actions are similar.

$$\mathcal{L}_r(\theta_e, \theta_{ds}, \theta_{do}) = \mathbb{E}[\sum_i (KL(f_{do}(z\_latent, o_i, a_i) || o_i'))$$
$$+ \lambda_1 \sum_i (f_{dr}(z\_latent, o_i, a_i) - r)^2] - \lambda_2 \sum_{j,j'} \cos(f_e(z_j), f_e(z_{j'})) \quad (2)$$

where $\lambda_1, \lambda_2$ are the scaling factors.

The third term in Equation (2) aims to maximize the cosine distance between any two skills, which encourages diverse policy learning for the skill learning module. The skill representation learning mechanism is first trained in the early exploration of our framework. After the early exploration is finished, we fix the parameters of the encoder in this module to generate representation features for all skill ids.

## 3.2 Skill Selector

Here, we introduce a skill selection mechanism that assigns each agent a skill. This procedure essentially assigns multiple skills to multiple agents. It is different from the single-agent skill-based RL problem, which only assigns multiple skills to one agent.

Relying only on observation features may result in the skill selector assigning the same skill to agents with similar observation features. Therefore, as shown in the middle part of Figure 1, the skill selector chooses skills based on latent observation features of agents and the latent skill variables generated in Section 3.1. The latent observation feature $o\_latent_i$ is obtained through an MLP and a GRU unit based on the observation $o_i$, which is similar to other MARL methods. Then $o\_latent_i$ is disposed by another MLP module, of which the output is multiplied by the latent skill variables $z\_latent_j, j = 1, 2, ..., M$. Then we can apply matrix multiplication of these two matrices, i.e., $o\_latent$ and $z\_latent$ and get the matrix of local Q-values $Q^\nu$ of all agents. We use $\Omega$ to represent the state space of the skill selector and $\Lambda$ to represent the action space. The Q-values of all agents' actions are denoted by $Q_1^\nu, Q_2^\nu, ..., Q_N^\nu$. In the execution mode, the index corresponds to the maximum value is the skill that should be assigned to the agent $i$. For the training of the skill selector, the learning objective is shown as Equation (3). Similar to the work in [21], a mixing network is introduced to compute $Q_{tot}^\nu$. Note that the assignment is computed every $k$ steps, thus the reward for each assignment is computed as $\sum_{t'=0}^{k-1} r_{t+t'}$. Moreover, in the early training stage, we encourage exploration and incentivize the skill selections to be as diverse as possible. The policy with high entropy forces the skill selector to explore distinguished states for agents and chooses different skills for agents, which benefits the training procedure. Therefore, maximizing the policy entropy is also a goal $\mathcal{G} = \mathcal{H}[Q_i^\nu | \Omega]$ in our objective. We replace it with an intrinsic reward $r^s = -\mathbb{E}_i[\log Q_i^\nu(o\_latent_i, z\_latent_i)]$ and add into Equation (3) for maximizing the expectation $\mathcal{G}$.

$$\mathcal{L}_s = [\sum_{t'=0}^{k-1} r_{t+t'} + \beta_s r^s + \gamma \max_{\Lambda'} Q_{tot}^{\nu-}(\Omega_{t+k}, \Lambda') - Q_{tot}^\nu(\Omega_t, \Lambda_t)]^2 \quad (3)$$

### 3.3 Skill-based Policy Learning

In this section, we introduce a skill learning mechanism, which consists of a skill discovery process and a skill-based policy learning process. The main idea of skill discovery includes two aspects. The first is that skills should do effective exploration in the environment, which means that different skills should explore effectively and access different states of the environment. Ideally, several skills should cover as much of the exploration space of the environment as possible. The second aspect is that skills should be learned as diverse as possible. In order to avoid duplication policies, each skill should compute the action that differs significantly from the others. For example, in cooperative football games, it is clear that the two skills of passing and shooting the ball should compute different actions so that the respective tasks can be completed.

We introduce mutual information from information theory to realize the aforementioned objectives. To achieve the first goal, we encourage each skill $z$ to associate with the observation $o$, i.e. maximizing the mutual information $\mathcal{I}(o; z)$ between skills and observations, where $\mathcal{I}$ represents the mutual information. As for the diversity of skills, we hope that different skills generate actions as differently as possible. Therefore, we can strengthen the correlation between the action $a$ and the skill $z$ through maximizing the mutual information $\mathcal{I}(a; z|o)$ conditioned on the given state $o$. Finally, we can improve the diversity of skills by maximizing the policy entropy $\mathcal{H}[a|o, z]$, where $\mathcal{H}[\cdot]$ is the Shannon entropy. In summary, the final maximization objective is shown as Equation (4):

$$
\begin{aligned}
r^m &= \mathcal{I}(o; z) + \mathcal{I}(a; z|o) + \mathcal{H}[a|o, z] \\
&= \underbrace{(\mathcal{H}[z] - \mathcal{H}[z|o])}_{\textcircled{1}} + \underbrace{(\mathcal{H}[a|o] - \mathcal{H}[a|o, z])}_{\textcircled{2}} + \underbrace{\mathcal{H}[a|o, z]}_{\textcircled{3}}
\end{aligned}
\tag{4}
$$

For term $\textcircled{2}$ in Equation (4), using the toolkit of variational inference and applying the bound from [1], we can construct the variational lower bound which is shown in Equation (5) based on **Theorem 1** in Appendix. (the proof is shown in Appendix A).

$$
\mathcal{H}[a|o] - \mathcal{H}[a|o, z] \geqslant \mathbb{E}_{p(a,z,o)}\left[\log \frac{p(z|a, o)}{p(a, z|o)}\right] - \mathcal{H}[a|o, z]
\tag{5}
$$

Further, we can transform $\mathbb{E}_{p(a,z,o)}\left[\log \frac{p(z|a,o)}{p(a,z|o)}\right]$ into Equation (6).

$$
-\mathbb{E}_{p(a,o)}\left[\mathcal{H}[p(z|a, o)]\right] + \mathbb{E}_{p(a,z,o)}\left[\log \frac{1}{p(z|o)}\right] + \mathbb{E}_{p(z,o)}\left[\mathcal{H}[p(a|z, o)]\right]
\tag{6}
$$

Inserting Equation (5) and (6) into Equation (4) yields Equation (7). Note that $\mathcal{H}[z|o]$ in term $\textcircled{1}$ and $\textcircled{3}$ are eliminated (see Appendix A for details).

$$
r^m \geqslant \mathcal{H}[p(z)] + \mathbb{E}_{p(z,o)}\left[\mathcal{H}[p(a|z, o)]\right] - \mathbb{E}_{p(a,o)}\left[\mathcal{H}[p(z|a, o)]\right]
\tag{7}
$$

$p(z)$ is the skill prior distribution. We fix this prior distribution to be uniform to ensure its maximization entropy inspired by [6]. To make full use of the information contained in all actions, inspired by [10], we use the Boltzmann Softmax distribution to generate a policy distribution, which is consistent with the original Q function with the greedy policy, i.e., $p(a|z, o) = \text{Softmax}(q_{\theta_a}(a|z, o))$. In this way, it takes into account all actions and makes maximum use of the information. Since it is difficult to exactly compute $p(z|a, o)$ covering all states, actions and skills, we make a variational approximation to $p(z|a, o)$ and optimize the lower bound of the variational posterior distribution $q_{\theta_z}(z|a, o)$.

The final optimization objective is shown as Equation (8).

$$
r^m \geqslant \mathbb{E}_{z,a,o}\left[\alpha_1 \log \text{Softmax}(q_{\theta_a}(a|z, o)) - \alpha_2 \log q_{\theta_z}(z|a, o)\right]
\tag{8}
$$

The discovered skills can be chosen by agents to perform skill-based policy learning. Each agent computes its skill-based policy $Q_i(a_i, o_i|z_j)$ where $z_j$ is the representation of the selected skill. To perform policy learning based on the global reward $r$ from the environment, we use a QMIX-style mixing network to mix all individual Q-values into a global Q-value $Q_{total}$. The TD-loss of the skill-based policy learning can be described as Equation (9).

$$
\mathcal{L}_{TD} = \mathbb{E}\left[\left(r + \gamma \max_{\mathbf{a}'} Q_{tot}^-(o', \mathbf{a}') - Q_{tot}(o, \mathbf{a})\right)^2\right]
\tag{9}
$$

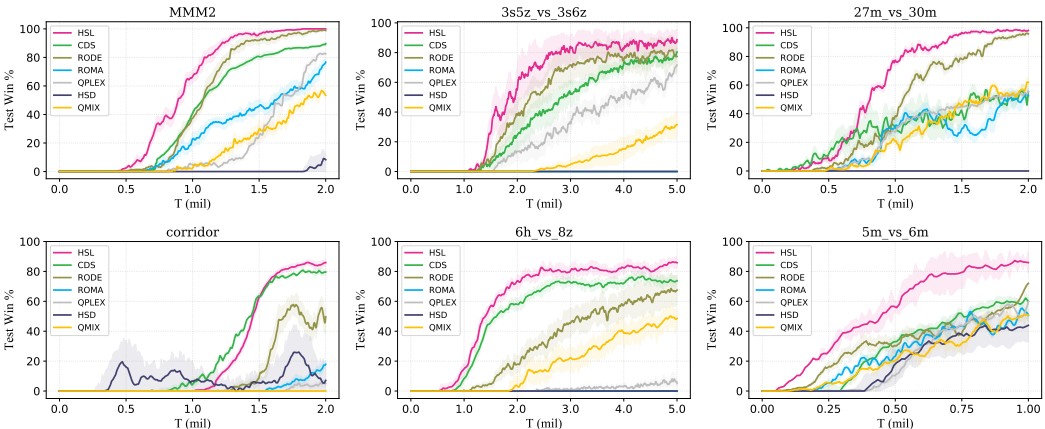

Figure 2: Performance comparison on SMAC.

### 3.4 Overall Optimization Objective

Finally, we describe how skill selector, skill discovery and skill-based policy learning are combined in our framework. It is organized as a bi-level learning structure.

Inspired by [6], the optimization goal $r^m$ of skill discovery is embedded as an intrinsic reward in $\mathcal{L}_p$ for skill-based policy learning, which is described as follows.

$$\mathcal{L}_p = \left[ r + \beta_m r^m + \gamma \max_{\mathbf{a}'} Q_{tot}^-(o', \mathbf{a}') - Q_{tot}(o, \mathbf{a}) \right]^2 \tag{10}$$

We train our framework end-to-end by optimizing the following objective function:

$$\mathcal{L} = \eta_s \mathcal{L}_s + \mathcal{L}_p \tag{11}$$

where $\mathcal{L}_s$ is the objective function for the skill selector and $\eta_s$ is the scaling factor in Section 3.2.

## 4 Experiments

In this section, we test the performance of our framework on three challenging multi-agent tasks, i.e., StarCraft II micromanagement multi-agent challenge (SMAC) [22], Google Research Football (GRF) [15] and GoBigger [8]. We compare our approach (HSL) with classical multi-agent value decomposition methods, i.e., QMIX [21] and QPLEX [26], role-based methods , i.e., ROMA[27],RODE[28], diversity-based method CDS [4] and skill-based method HSD [32]. All experiments are conducted over five random seeds. The detailed setting of the experimental setup is described in Appendix B.2.

### 4.1 Performance

**SMAC**: SMAC is a cooperative multi-agent task, in which each agent cooperates with teammates to kill enemies. Maps in SMAC can be classified as Easy, Hard and Super Hard. In this section, we concentrate on one Hard scenario *5m_vs_6m* and all Super Hard scenarios, i.e., *MMM2, 3s5z_vs_3s6z, 27m_vs_30m, corridor, 6h_vs_8z*.

Figure 2 shows comparing results of the proposed method, HSL, with other baseline methods. As we can see, HSL achieves the best performance over all scenarios. In the first three scenarios, even if there exists a baseline that could achieve a satisfactory win rate, HSL still has a much faster convergent speed. In the last three scenarios, HSL obtains the best win rate. Especially for $6h\_vs\_8z$ and $5m\_vs\_6m$, HSL improves the win rate by around $10\%$ and $20\%$ respectively, compared to CDS.

It is interesting to see that CDS does not perform well in the scenario $27m\_vs\_30m$. It is consistent with the explanation in the introduction that CDS is not suited for large-scale scenarios. CDS introduces extra neural networks for each agent. All these extra neural networks do not share

parameters in order to ensure diversity. Although HSL also introduces one extra neural network for learning skills, different skill representations ensure diversity and thus parameter sharing still could be applied.

Another note is that the role-based MARL, i.e., RODE, performs better than CDS in the first two scenarios. The reason is that agents in the two scenarios are heterogeneous, which is easy for RODE to distinguish roles based on observations and actions of agents. HSL is still better than RODE by introducing skills. For the last three scenarios in which agents are homogeneous, it is difficult for RODE to distinguish roles. HSL could obtain the best performance by discovering diverse skills. In addition, we can observe that HSD cannot win in Super Hard scenarios. The skill discovery mechanism based only on each agent's local observation makes it difficult to learn winning strategies effectively.

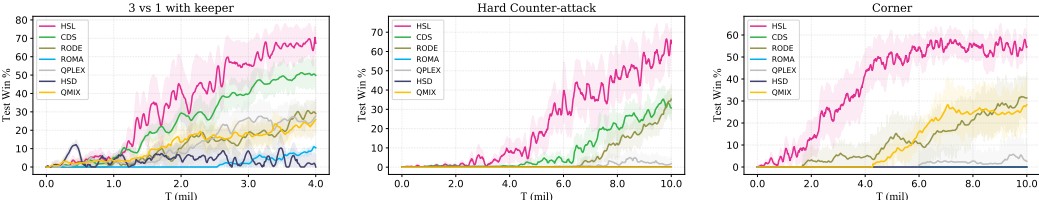

Figure 3: Performance comparison on GRF.

**GRF**: In GRF, agents work with each other to decide when and where to move and keep the ball as long as possible to maximize the chances of scoring. We consider three scenarios, i.e., *3 vs 1 with keeper, Hard Counter-attack* and *Corner*.

Results are shown in Figure 3, and again HSL performs the best. It is clear to see that classical value decomposition MARL methods, i.e., QMIX and QPLEX, perform poorly in these scenarios due to the lack of disposing of heterogeneous tasks. Role-based MARL methods cannot achieve satisfactory performance because agents in these scenarios usually get similar states and actions, making it hard to distinguish roles for agents. Due to the effective skill representation and skill selector mechanism, HSL performs better than role-based MARL methods. Further, we could see that CDS is not able to find effective strategies to win on *Corner*. Still, the reason is that the number of agents in this scenario is much higher than that in the other two scenarios, which leads to an adverse effect on training extra networks in CDS.

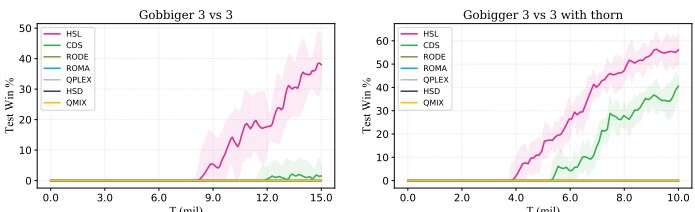

Figure 4: Performance comparison on GoBigger.

**GoBigger**: In GoBigger tasks, agents control one or more circular balls and cooperate to become large as much as possible by eating food balls and other smaller balls in the environment. Meanwhile, each agent needs to prevent itself from being eaten by other larger enemy balls. Agents can move, eject, split and stop, which could be discretized as 13 actions, including moving, ejecting and splitting in four orthogonal directions and stopping. For each episode, the team with the largest overall size will win at the end of the game.

We show the performance comparison in Figure 4. We can observe that these two maps in GoBigger are quite difficult. Only HSL and CDS can learn winning policies. Role-based MARL methods do not work at all due to the reason explained before. In GoBigger, it becomes quite hard for role-based MARL methods to find roles based on observations and actions. Due to the effective skill representation and skill assignment, agents in HSL can select useful skills to cooperate with each other to eat more food and eliminate enemies.

## 4.2 Ablation Study

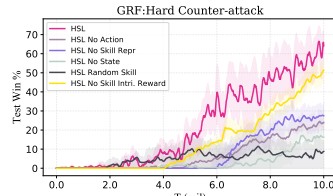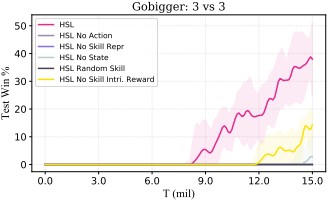

Figure 5: Ablation studies.

In this section, we conduct ablation experiments to evaluate the influence of the three components: (A) Skill representation learning; (B) Skill selector; (C) Skill-based policy learning. To test component A, we replace representations of skills with raw states, which is denoted by *HSL No Skill Repr.* For evaluating component B, we introduce *HSL Random Skill*, where each agent randomly selects a skill. Also, we remove the intrinsic reward in the learning objective of the skill selector, which is associated with *HSL No Skill Intri. Reward.* To test component C, we consider *HSL No State* which removes term ①, and *HSL No Action* which removes term ② and ③ in Equation (4).

Results are shown in Figure 5. *HSL Random Skill* obtains the worst performance, which indicates that the skill selector plays a key role in the performance of HSL. Compared with *HSL Random Skill*, *HSL No Skill Repr.* achieves higher performance in *SMAC:6h_vs_8z* and *GRF:Hard Counter-attack*, since the raw state of agents could also work for skill assignment. However, it still performs worse than our skill representation technique. It becomes obvious in the task *Gobigger:3_vs_3*. SM No Skill Repr. does not work at all. *HSL No Action* achieves higher performance than *HSL No State* does in *SMAC:6h_vs_8z* and *GRF:Hard Counter-attack*, which shows that the term ① in (4) is important in these scenarios. In contrast, other terms in (4) are important in *HSL No State* because the performance is higher in this scenario. As for the intrinsic reward in the skill selector, we can observe that the performance gap between HSL and *HSL No Skill Intri. Reward* in *GRF:Hard Counter-attack* and *Gobigger:3_vs_3* is larger than that in *SMAC:6h_vs_8z*. The reason is that dense rewards in *SMAC:6h_vs_8z* help the skill selector learn effective skill selection policies. Conversely, the sparse reward in *GRF:Hard Counter-attack* and the inefficient reward in *Gobigger:3_vs_3* increase the policy learning difficulty of the skill selector in *HSL No Skill Intri. Reward*.

## 4.3 Skill Demonstration

In this section, we conduct a case study on scenario *SMAC: 6h_vs_8z* to demonstrate what our skill-based framework learns. We carry out an experiment with the obtained strategy and give a skill demonstration in Figure 6. It is clear to see that our framework has learned 3 skills including escaping (red), attacking (green) and kiting (yellow). The middle part shows the skill selection of agents during the game. For example, Agent 1 selects the red skill at the beginning and the yellow skill until dies. The top and the bottom part give visualizations of discovered skills.

In the first two graphs, two agents select the escaping skill to absorb damage and other agents select the attack skill to produce damage to enemies. In the third graph, one agent close to enemies selects escaping skill. Four other agents select the kiting skill to run in the opposite direction to make damage to enemies. In the fourth graph and fifth graph, two of the alive agents still choose the kiting skill. Agent 3 first chooses the attacking skill since it is far away from enemies, and soon it chooses to escape when enemies are approaching. In the last graph, an escaped agent is died, while the remaining agents select the kiting skill to eliminate enemies. Results on GRF and GoBigger benchmarks are detailed in Appendix B.3.

## 5 Related Work

Deep MARL algorithms have drawn broad attention in recent years. Most works take advantage of the CTDE paradigm [9] for cooperative policy learning for agents. Policy-based MARL algorithms design a centralized critic to compute gradients for decentralized actors such as MADDPG[16], COMA [7] and DOP [30]. Value-based MARL algorithms achieve centralized optimization and

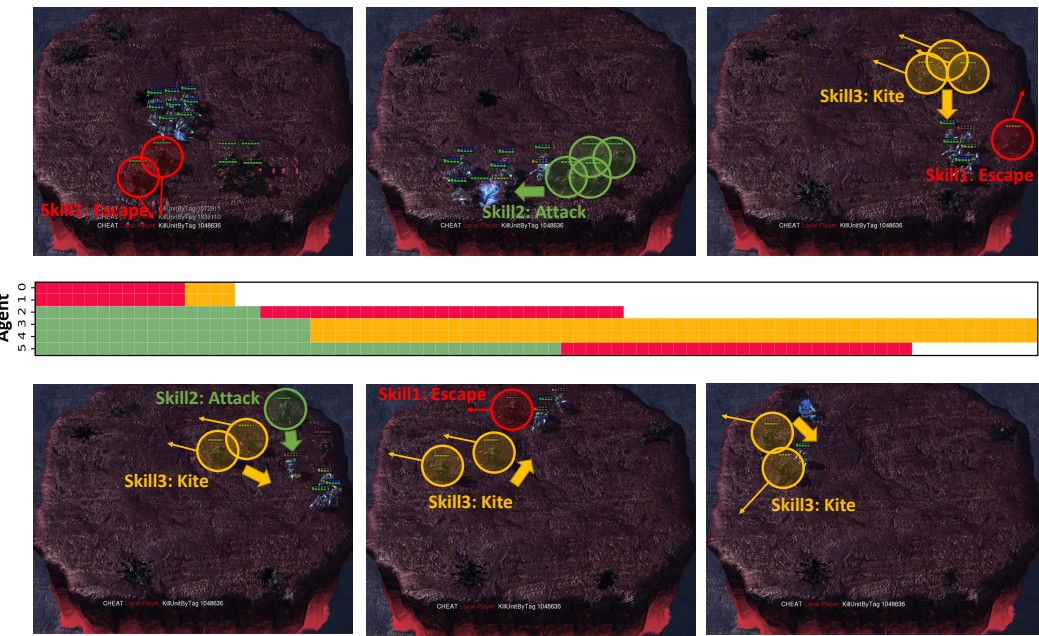

Figure 6: Skill demonstration on *SMAC: 6h_vs_8z.*

decentralized execution via value decomposition. These methods introduce a mixing network to mix the local value functions of each agent to estimate the joint value function. VDN [25], QMIX [21] and QTRAN [23] have gradually improved the decomposition capabilities of the mixing networks. Qatten [34] refines the representation of the joint value function by implementing a multi-head attention formation to approximate the value decomposition procedure. Qplex [26] designs a duplex dueling network to encode the IGM principle for higher representation capabilities of the IGM class.

Although many MARL works under the CTDE paradigm have achieved promising results, most of them apply the parameter sharing technique to share knowledge and experience among agents to compute cooperative actions. However, when faced with complex multi-agent tasks, centralized training and parameter sharing lead to difficulties in generating heterogeneous behavior. Therefore, a series of works have been developed to learn heterogeneous policy learning in MARL. ROMA [27] introduces the concept of role to compute independent actions for different agents. RODE [28] decomposes the action space into sub-groups in order to reveal proper roles for agents for heterogeneous policy learning. Recent works introduce diversity to enable agents to learn diverse policies to solve complex tasks. MAVEN [17] designs a latent space to learn heterogeneous policies and encourage exploration. EITI and EDIT [29] use mutual information to encourage each agent to perform independent exploration to optimize team performance. EMC [35] proposes a curiosity-driven exploration method to make full use of the explored valuable experiences for each agent's efficient policy learning. EOI [12] learns each agent's policy via the combination of the gradient from the intrinsic value function and the joint value function. CDS [4] adds an extra local policy network for each agent and tries to make a balance between the high learning efficiency brought by diversity and parameter sharing. The most relevant work is HSD [32], which encourages skill discovery in MARL settings. It proposes a bi-level hierarchy to discover cooperative skills for each agent. However, HSD selects skills for agents only based on historical observations, which leads to poorly distinguishable skill discovery. Skill-based policy learning mechanism in HSD cannot learn diverse skills because it is also conditioned on skill features. By comparison, our approach does not totally rely on trajectories, but instead designs a skill representation mechanism and introduces intrinsic rewards for skill selection. Furthermore, our approach computes intrinsic targets for skill discovery and skill-based policy learning via mutual information maximization.

# 6 Conclusion

This paper proposes a novel skill-based MARL framework to deal with the heterogeneous problem in complex cooperative multi-agent tasks. The concept of skill is introduced, which is represented as latent variables generated by an auto-encoder model. The representation reflects the characters of agents and ensures the diversity of skills, which play an important role in generating and selecting skills for agents. The skill selection mechanism outputs a skill for an agent based on its observation and skill representation. The selection is dynamically changed with the real-time game state, which helps agents to use proper skills to adapt to complex situations. To ensure efficient policy learning, we propose a skill-based policy learning mechanism based on mutual information optimization under the information theory paradigm. Empirical results on three challenging MARL benchmarks show that our framework significantly pushes forward the performance of state-of-the-art MARL methods. We expect that our approach could provide valuable guidance for future MARL in using skills to solve complex cooperative problems in the real world.

# 7 Acknowledge

This work was partially supported by the National Natural Science Foundation of China (No.61902425) and the Open Fund of Science and Technology on Parallel and Distributed Processing Laboratory (WDZC20205500104). We would like to thank the anonymous reviewers for their valuable suggestions.

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
