## A  Mathematical Derivation

The intrinsic reward function of skill discovery is shown as follows.

$$r^m = \mathcal{I}(z;s) + \mathcal{I}(a;z|s) + \mathcal{H}[a|s,z]$$
$$= \underbrace{(\mathcal{H}[z] - \mathcal{H}[z|s])}_{\textcircled{1}} + \underbrace{(\mathcal{H}[a|s] - \mathcal{H}[a|s,z])}_{\textcircled{2}} + \underbrace{\mathcal{H}[a|s,z]}_{\textcircled{3}} \qquad (12)$$

We first analysis $\mathcal{H}[a|s]$ in term $\textcircled{2}$. We present a lower bound of the policy entropy $\mathcal{H}[a|s]$. Here, we give the details of its proof as follows.

**Theorem 1.** *The lower bound on the policy entropy $\mathcal{H}[a|s]$ corresponds to:*

$$\mathcal{H}[a|s] \geqslant \mathbb{E}_{p(a,z,s)}\left[\log\frac{p(z|a,s)}{p(a,z|s)}\right] \qquad (13)$$

*Proof.*

$$\mathcal{H}[a|s] = -\sum_{s'} p(s')H[a|s=s']$$
$$= \sum_{s'} p(s') \sum_{a'} p(a'|s') \log\frac{1}{p(a'|s')}$$
$$= \sum_{s'} \sum_{a'} p(a',s') \log\frac{1}{p(a'|s')}$$
$$= \sum_{s'} \sum_{a'} p(a',s') \log\left[\sum_{z'} p(z'|a',s')\frac{1}{p(a'|s')}\right]$$
$$= \sum_{s'} \sum_{a'} p(a',s') \log\left[\sum_{z'} p(z'|a',s')\frac{p(z'|a',s')}{p(a',z'|s')}\right]$$
$$\geqslant \sum_{s'} \sum_{a'} p(a',s') \sum_{z'} p(z'|a',s') \log\left[\frac{p(z'|a',s')}{p(a',z'|s')}\right]$$
$$= \sum_{s'} \sum_{a'} \sum_{z'} p(a',z',s') \log\left[\frac{p(z'|a',s')}{p(a',z'|s')}\right]$$
$$= \mathbb{E}_{p(a,z,s)}\left[\log\frac{p(z|a,s)}{p(a,z|s)}\right] \qquad (14)$$

$\square$

Then we describe how to insert Equation 5 and 6 into Equation 4 to get Equation 7.

Equation 15 details how to transform $\mathbb{E}_{p(a,z,s)}\left[\log\frac{p(z|a,s)}{p(a,z|s)}\right]$ into Equation 6.

$$\mathbb{E}_{p(a,z,s)}\left[\log\frac{p(z|a,s)}{p(a,z|s)}\right] = \mathbb{E}_{p(a,z,s)}[\log p(z|a,s)] + \mathbb{E}_{p(a,z,s)}\left[\log\frac{1}{p(a,z|s)}\right]$$

$$= \mathbb{E}_{p(a,s)}[\mathbb{E}_{p(z|a,s)}\log p(z|a,s)] + \mathbb{E}_{p(a,z,s)}\left[\log\frac{1}{p(z|s)p(a|z,s)}\right]$$

$$= -\mathbb{E}_{p(a,s)}\left[\mathbb{E}_{p(z|a,s)}\frac{1}{\log p(z|a,s)}\right] + \mathbb{E}_{p(a,z,s)}\left[\log\frac{1}{p(z|s)} + \log\frac{1}{p(a|z,s)}\right]$$

$$= -\mathbb{E}_{p(a,s)}[\mathcal{H}[p(z|a,s)]] + \mathbb{E}_{p(a,z,s)}\left[\log\frac{1}{p(z|s)}\right] +$$

$$+ \mathbb{E}_{p(z,s)}\left[\mathbb{E}_{p(a|z,s)}\left[\log\frac{1}{p(a|z,s)}\right]\right]$$

$$= -\mathbb{E}_{p(a,s)}[\mathcal{H}[p(z|a,s)]] + \mathbb{E}_{p(a,z,s)}\left[\log\frac{1}{p(z|s)}\right] + \mathbb{E}_{p(z,s)}[\mathcal{H}[p(a|z,s)]] \tag{15}$$

For the term $\mathbb{E}_{p(a,z,s)}\left[\log\frac{1}{p(z|s)}\right]$, we can transform it into:

$$\mathbb{E}_{p(a,z,s)}\left[\log\frac{1}{p(z|s)}\right] = \sum_s\sum_a\sum_z p(a,z,s)\log\left[\frac{1}{p(z|s)}\right]$$

$$= \sum_s\sum_z\sum_a p(a)p(z,s|a)\log\left[\frac{1}{p(z|s)}\right]$$

$$= \sum_s\sum_z p(z,s)\log\left[\frac{1}{p(z|s)}\right]$$

$$= H[z|s] \tag{16}$$

Equation 17 details how to get Equation 7.

$$r^m = \mathcal{I}(z;s) + \mathcal{I}(a;z|s) + \mathcal{H}[a|z,s]$$

$$= (\mathcal{H}[z] - \mathcal{H}[z|s]) + (\mathcal{H}[a|s] - \mathcal{H}[a|z,s]) + \mathcal{H}[a|z,s]$$

$$\geqslant (\mathcal{H}[z] - \mathcal{H}[z|s]) + \mathbb{E}_{p(a,z,s)}\left[\log\frac{p(z|a,s)}{p(a,z|s)}\right]$$

$$= (\mathcal{H}[z] - \mathcal{H}[z|s]) + \left(-\mathbb{E}_{p(a,s)}[\mathcal{H}[p(z|a,s)]] + \mathbb{E}_{p(a,z,s)}\left[\log\frac{1}{p(z|s)}\right] + \mathbb{E}_{p(z,s)}[\mathcal{H}[p(a|z,s)]]\right)$$

$$= \mathcal{H}[p(z)] + \mathbb{E}_{p(z,s)}[\mathcal{H}[p(a|z,s)]] - \mathbb{E}_{p(a,s)}[\mathcal{H}[p(z|a,s)]] \tag{17}$$

## B Experimental Details

### B.1 Baselines

We compare our approach with multi-agent value decomposition methods (QMIX and QPLEX), role-based methods (ROMA and RODE), diversity-based method (CDS) and skill-based method (HSD). For QMIX, QPLEX, RODE, ROMA, CDS and HSD, we use the codes with fine-tuned hyper-parameters, which are provided by the authors.

### B.2 Architecture and Hyper-parameters

We develop our method based on the Python MARL framework (PyMARL) on the github. Configuration of the hyper-parameters of the agent network and the mixing network are the same as those in QMIX, which could be found in the source codes. We list these hyper-parameters of our method in

Table 1: Hyper-parameters for HSL.

| Parameter | Value |
|---|---|
| *Algorithm hyper-parameters* | |
| Discount factor | 0.99 |
| Batch size | 32 |
| Buffer size | 5000 |
| Optimizer | RMSprop |
| Learning rate | 0.0005 |
| Interval of target network update | 200 |
| *Agent network hyper-parameters* | |
| Temporal module in Agent network | GRU |
| Dimensions of hidden states of temporal module | 64 |
| *Mixing network hyper-parameters* | |
| Dimensions of mixing network embedding | 32 |
| Number of hyper network layers | 2 |
| Dimensions of hyper network embedding | 64 |
| *HSL hyper-parameters* | |
| Dimensions of skill representation encoder embedding | 20 |
| Reward decoder scaling factor | 10 |
| Cosine distance scaling factor | 0.1, 1 |
| Skill representation learning mechanism training steps | 50000 |
| Dimensions of skill selector encoding network embedding | 32 |
| Decision interval of the skill selector | 5 |

the first three parts in Table 1. For exploration, we use $\epsilon$-greedy in the training procedure and keep $\epsilon$ constant for the test. The $\epsilon$ anneal times are different on different environments. In most of the scenarios in SMAC and all scenarios in GRF, $\epsilon$ is annealed linearly from $1.0$ to $0.05$ over $50K$ time steps. For Super Hard scenarios *3s5z_vs_3s6z*, *6h_vs_8z* in SMAC and all scenarios in GoBigger, $\epsilon$ is annealed linearly from $1.0$ to $0.05$ over $500K$ time steps. Configuration of the hyper-parameters of the skill representation learning mechanism and the skill selector in our framework is shown in the last part in Table 1. Particularly, the cosine distance scaling factor is set to 1 in all scenarios in SMAC and 0.1 in all scenarios in GRF and GoBigger.

For configuration of the rest hyper-parameters in our framework, we list them in Table 2. We first introduce the important hyper-parameter of the number of the skills. In fact, the number of skills is dependent on the difficulty of one scenario and the number of agents in this scenario. Then we introduce hyper-parameters related to the intrinsic reward, i.e., $\beta_s, \beta_m, \alpha_1, \alpha_2$. In SMAC, $\beta_s$ is set to $0.01$. In GRF and GoBigger, $\beta_s$ is set to $0.1$ because skill selector plays an important role in these scenarios providing similar observations for agents. For $\beta_m, \alpha_1, \alpha_2$, we apply the grid search mechanism ($\beta_m, \alpha_1, \alpha_2 \in \{-1, -0.1, -0.01, 0.01, 0.1, 1, 2\}$) to fine-tune these hyper-parameters. All the tuned hyper-parameters are shown in Table 2.

## B.3 MARL benchmarks

**SMAC**: Each agent receives local observation based on its field of view at each timestep. The local observation includes information about the map within a circular area whose radius equals the sight range(9) around each agent. The feature vector of the local observation consists of 4 parts: available movements, enemies' attributes, allies' attributes and attributes of the agent itself. Available movements have 4 bits, and each bit indicates whether the agent can move in this direction or not. Attributes of enemies and allies are similar, which contain information about *distance, relative x, relative y, health, shield and unit_type*. Attributes of the agent itself have *health, shield and unit_type*.

The global state contains information about all units on the map. The state vector includes features present in agents' observations. Particularly, the state vector contains the *coordinates* of all agents

Table 2: Hyper-parameters for the skill-based policy learning mechanism in HSL.

| | Environment | $n_{skill}$ | $\beta_s$ | $\beta_m$ | $\alpha_1$ | $\alpha_2$ |
|---|---|---|---|---|---|---|
| SMAC | MMM2 | 4 | 0.01 | 0.1 | 1 | -0.1 |
| | 3s5z_vs_3s6z | 4 | 0.01 | 0.01 | 1 | -0.1 |
| | 27m_vs_30m | 3 | 0.01 | 0.1 | 1 | -0.1 |
| | corridor | 3 | 0.01 | 1 | 1 | 0.1 |
| | 6h_vs_8z | 3 | 0.01 | 0.1 | 1 | -0.1 |
| | 5m_vs_6m | 3 | 0.01 | 0.1 | 1 | -0.1 |
| GRF | 3 vs 1 with keeper | 3 | 0.1 | 1 | 1 | 1 |
| | Hard Counter-attack | 4 | 0.1 | 0.1 | 1 | 0.1 |
| | Corner | 5 | 0.1 | 1 | 1 | 0.01 |
| GoBigger | GoBigger 3 vs 3 | 3 | 0.1 | 1 | 0.1 | 1 |
| | GoBigger 3 vs 3 with thorn | 3 | 0.1 | 1 | 0.1 | 0.1 |

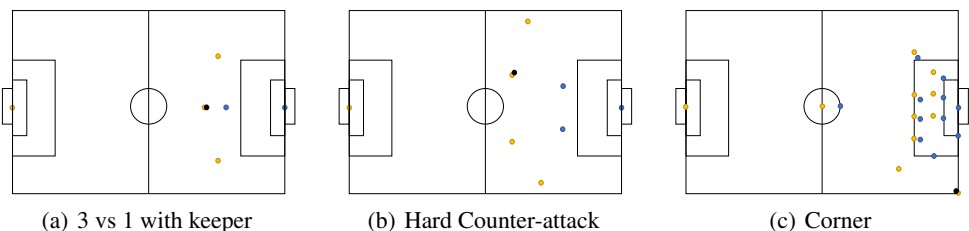

(a) 3 vs 1 with keeper      (b) Hard Counter-attack      (c) Corner

Figure 1: Visualization of three scenarios in Google Research Football.

relative to the center of the map. Moreover, *energy* of the special unit Medivacs, *cooldown* of the rest units and the last actions of all agents are also included.

Actions of agents belong to a discrete action space. Each agent can select an action from the following: *move north, south, east and west, attack one of the enemies, stop and the null action*. Notice that the number of attack actions in the attack action set equals the number of enemies. The Medivacs, which is a healer unit, uses *heal* actions instead of *attack* actions. The maximum number of actions ranges from 7 to 70, which depends on the scenario.

Rewards received by agents consist of total damage to enemies, points for killing opponents and scores for winning the game.

Enemy units are controlled by a built-in handcrafted AI. The game ends when all units on one side die or the time exceeds a fixed period.

**GRF**: In GRF tasks, agents are supposed to cooperate in timing and positions to seize the fleeting opportunity to score goals. MARL controls left-side players (in yellow in the visualization). The right-side players are controlled by rules from the game engine. Notice that goalkeepers of both sides are also controlled by rules.

The observation of each agent consists of 3 parts, namely the positions of the ego-agent, other-agents and the ball. This local observation calculates relative positions between the agent itself and other objects. In addition, the moving directions of these objects are also included in the observation.

Similarly, the global state contains the absolute positions and the moving directions of all players and the ball. The z-coordinate of the ball is also included

Agents have a discrete action space of 19. Each agent can select an action from moving in eight directions, passing, sliding, and shooting.

Rewards received by agents are only related to scoring. Agents can only score to get positive rewards. The game ends when agents score and get rewards.

The detailed configuration of each scenario in our experiments is shown in Figure 1.

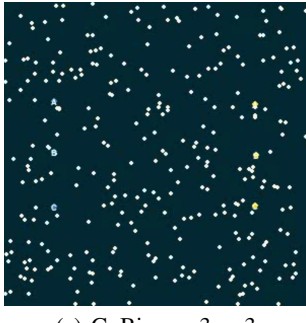 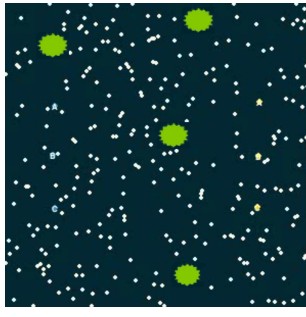

(a) GoBigger: 3 vs 3           (b) GoBigger: 3 vs 3 with thorn

Figure 2: Visualization of two scenarios in GoBigger.

**GoBigger**: GoBigger is a multi-agent decision intelligence environment. Players control one or multiple balls in GoBigger to gain as much size as possible by eating other smaller balls in the environment. Each player can choose operations provided by GoBigger such as moving, splitting, ejecting and stopping. Moving means balls can move in a certain direction with a custom speed and acceleration. Ejecting means balls can eject a spore-ball to decrease its size and make itself move faster. Splitting allows balls to split themselves into two pieces of the same size. Stopping first stops the balls and then gather all balls slowly if it has several clone balls. In GoBigger, there are 4 kinds of balls, i.e., the food ball, the thorn ball, the spore ball and the player ball. The player ball is controlled by the player. Food balls are the neutral resources which are static in the game. A player ball can gain its size by eating food balls. If the player ball eats a thorn ball whose size is smaller, the player ball will be split into several pieces.

In our experiment, we add two teams to the game. One of the teams is controlled by our framework and the other is controlled by the built-in rule in GoBigger. For both scenarios used in our experiments, the map size is $500 \times 500$ and the size of the player ball's partial vision is 50. There are 300 food balls in these scenarios. Particularly, the scenario *GoBigger 3 vs 3 with thorn* has 4 thorn balls.

The local observation of each agent contains 5 parts. The first is the coordinates of the partially observed rectangle of each agent. The second is the features of food balls. Features of food balls are constructed as a grid. Each grid computes the density of food balls in this grid and the offsets of its coordinates compared with those of the player ball. The next part is the relative positions and radius of thorn balls. Then comes the relative positions of spore balls. The final part is the relative positions, radius, the player id and the team id of other player balls containing allies and enemies.

The global state is similar to the local observation of each player ball except for the positions and the features of food balls. The global state computes the absolute positions of all objects. Features of food balls are still absolute positions instead of the density grid.

GoBigger has a hybrid action space. For simplicity, we design discrete action space to split into 4 directions. Agents can move, eject and split into 4 directions. With the stop action, the size of the discrete action space is 13.

Rewards received by agents are related to the size increase of the controlled player ball.

If one of the player ball is eliminated by other player balls, it will be respawned. The game ends when the maximum time limit is reached.

The detailed configuration of each scenario in our experiments is shown in Figure 2.

## B.4 Infrastructure

A PC with CPU Intel Core i9-10920X and GPU NVIDIA RTX 8000 is utilized to run all experiments. The training time ranges from 6 hours to 60 hours, which is based on the number of allied agents and enemies of each scenario, the map size and the total training steps.

# C  Additional Results

In this section, we show results of additional experiments. These experiments include skill demonstrations on SMAC benchmark and GRF benchmark.

## C.1  Skill Demonstration on GRF

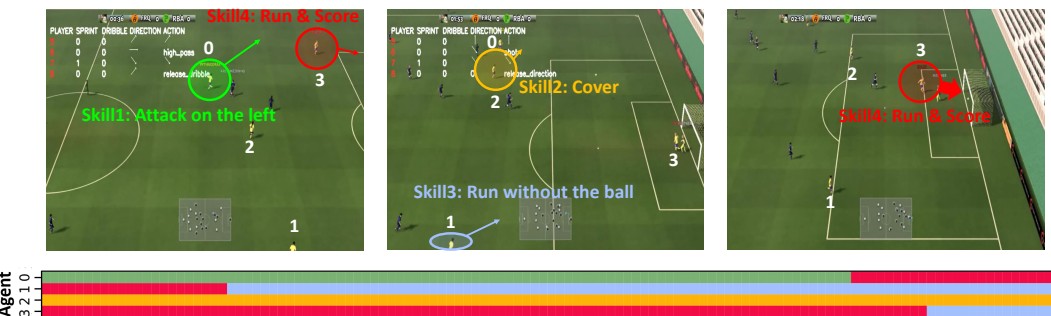

Figure 3: Skill demonstration on *GRF: Hard Counter-attack*.

We then describe experimental results of skill demonstration on GRF benchmark. We choose the *Hard Counter-attack* scenario and train HSL with 4 skills in order to visualize the skill demonstration intuitively. Results are shown in Figure 3. Figure 3 contains skill assignments at the bottom part and visualization of discovered skills at both the top part. Our framework has discovered 4 skills, i.e., attacking the left, covering, running without the ball, running and scoring. We describe the skill assignment and skill-based policies in order from left to right of Figure 3. In the first graph, agent 0 keeps the ball and chooses the skill of attacking the left. Agent 3, choosing the skill of running and scoring, cooperates closely with agent 0. By selecting this skill, agent 3 runs as close to the goal as possible and seizes the opportunity to score. In the second graph, agent 2 chooses the covering skill to cooperate with agent 0 who keeps the ball. In fact, agent 2 chooses this skill through the match. Agent 1 chooses the skill of running without the ball. In the last graph, agent 0 passes the ball to agent 3. Agent 3 with the skill of scoring can score once it gets the ball.

## C.2  Skill Demonstration on GoBigger

In this section, we conduct a case study on scenario *Gobigger: 3_vs_3 with thorn* to demonstrate what HSL learn, which is shown in Figure 4.

In the figure, the horizontal arrow shows the time step of this experiment, while the line chart shows the total size of agents in our team changes over time. Results of the skill assignment and skill-based policies are shown at the top and the bottom of the figure, respectively. We can observe that HSL has learned three skills, i.e., collision with thorn balls, eating food balls and eliminating enemy balls. Based on the change in the size of the agents, we take three parts to illustrate different skills. In the first part, the skill selector chooses the first skill for the agent near the thorn ball. This agent soon collides with the thorn ball and splits into many clones, resulting in a rapid increase in total size later. Then most agents are assigned the second skill, which is to find food balls and eat them to get most of the reward. In the last part, one agent meets an enemy ball which is much smaller. The skill selector chooses the last skill for the agent, allowing the agent to split in the direction of the enemy ball and eliminate it.

# D  Limitations

The limitations of our proposed HSL are three-fold. 1) Rewards for the skill selector are not proper enough for efficient training. In this paper, we directly use rewards from the environment to train the skill selector. However, the skill selection policy is independent of the agents' policy, which requires an explicitly designed skill selection reward design. In particular, training of the skill selector in multi-agent tasks with sparse rewards can benefit a lot from independently

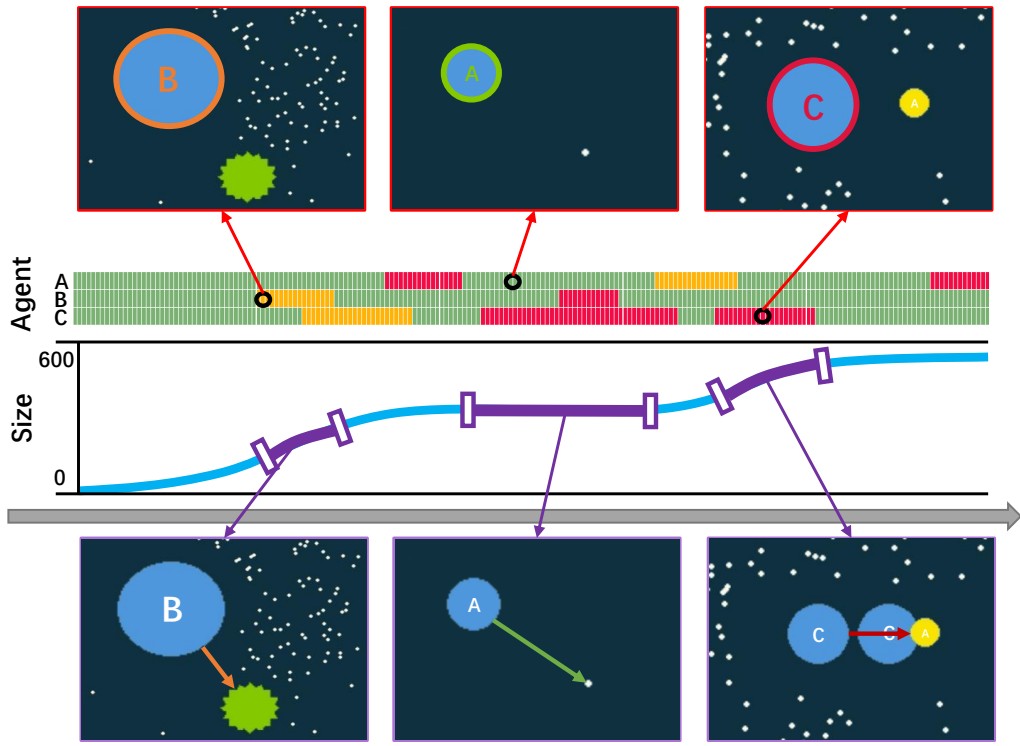

Figure 4: Skill demonstration on *GoBigger: 3 vs 3 with thorn*.

designed reward mechanisms. Although an intrinsic reward based on the policy entropy in HSL encourages diverse skill selection, the gap between it and the ideally designed reward mechanism is still huge. 2) There is no explicit design for a mechanism for synchronous course of the skill execution, which might introduce side effects to the training efficiency. The ideal way is to train skill policies conditioned on all $z$-ids simultaneously. Without redevelopment of multi-agent environments, we can only assign specific skills for agents to achieve this goal coarsely.

3) HSL is challenged by huge numbers of skills in complex multi-agent tasks. In our experiment, we set the number of skills to be less than or equal to the number of agents in MARL benchmarks, which is proved to achieve satisfactory performance. Multi-agent tasks in the real world usually contain many agents and rely on numerous skills. In fact, we have conducted experiments on setting more skills in our framework. Results are shown in Figure 5. However, we find our framework with more skills cannot achieve better performance than that with the number of skills less than or equal to the number of agents.

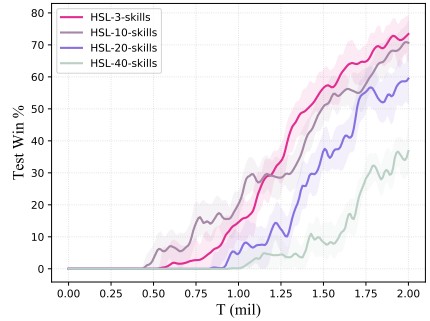

Figure 5: Comparison of HSL with different numbers of skills on *6h_vs_8z*.

## E Broader impact

We propose a novel multi-agent reinforcement learning method that leverages the skill mechanism to learn heterogeneous policies for agents. Our work can contribute to a wide range of applications, including multi-agent games, robotics and quantitative finance. For example, in Multiplayer Online Battle Arena (MOBA) games, the MARL method usually controls five different heroes to fight with human players. Our work enables each agent to learn many useful skills to cooperate with each other efficiently and thus beat the top human players. Also, in quantitative finance scenarios,

our method can help agents learn valuable heterogeneous trading policies. The goal of our method is to tackle a fundamental problem in multi-agent reinforcement learning. Therefore, we do not anticipate a direct negative outcome. In practical applications involving our method, potentially negative outcomes might occur.

Here we give two examples to show the potential negative outcome of our method in practical applications. The first is related to the human oversight of a MARL system. Trusty AI systems must allow for human oversight to support human autonomy and decision-making. However, MARL applications may pose challenges to human oversight because these applications aim to increase the autonomy of machines. For example, a MARL system for monitoring and adjusting energy usage in a building may be constantly making so many small decisions. These decisions are difficult for a human to review and change decisions after the fact. One way to address this problem is to impose constraints while the system is being designed such as waning a human if levels go beyond a certain threshold. The second is related to the security of a MARL system. For example, even a demonstrably safe MARL-based multi robot system could be forced into dangerous collision scenarios by perturbing its sensory input or disrupting its reward function. Possible ways to address is to oversee the transparency of the training data and the reward function or to develop safe multi-agent reinforcement learning methods.

## F    Discussion

First, we talk about the potential to improve the training efficiency of our framework. Our framework contains a skill representation learning mechanism, a skill selector and a skill-based policy learning mechanism. The skill representation learning mechanism is first trained in the early exploration of our framework. After the early exploration is finished (usually 50000 time steps), the training of the skill representation learning mechanism stops. We fix the parameters of the encoder in this module to generate representation features for all skill ids. The skill selector and the skill-based policy learning mechanism are organized as a bi-level learning structure. Although we get satisfactory results on challenging MARL benchmarks, we can further improve the training efficiency of our framework. In this paper, we use Q-learning method to train the skill selector. However, based on the efficient skill representation mechanism, we reckon that it can be replaced with a more lightweight approach, such as dot production.

Then comes the future study. Improving the quality of skill representations is a straightforward direction. We can design a more flexible and lightweight skill selector. For example, it is interesting to introduce GCN-based graph clustering methods in the skill selector. Besides, designing a particular mechanism to perform skill-based policy learning with more skills in our framework is also promising.