# OpenReview forum: "Heterogeneous Skill Learning for Multi-agent Tasks"
_NeurIPS.cc/2022/Conference — NeurIPS 2022 Accept_

### Official Review · Reviewer_amao · 2022-07-09

**Rating:** 6
**Confidence:** 3
**Soundness:** 3 good
**Presentation:** 3 good
**Contribution:** 3 good

**Summary:**

The authors proposed a skill-based MARL framework to enable agents to master diverse skills. The framework consists of the skill representation mechanism with an auto-encoder model, the skill selector to realize the assignment from agents to skills, and the skill-based policy learning mechanism with a mutual information-based intrinsic reward function. Experimental results show that the framework obtains the best performance on three challenging benchmarks, i.e., StarCraft II micromanagement tasks, Google Research Football and GoBigger, over state-of-the-art MARL methods.


**Questions:**

* L45- “However, extra introduced networks for all agents hinder the application of CDS on large-scale tasks, which will be illustrated in the experiment.” The proposed HSL requires extra networks to obtain the skill diversity. Since HSL outperformed CDS in MARL performance, the problem of CDS can be mentioned differently.
* L248- In association with the above, “Still, the reason is that the number of agents in this scenario is much higher than that in the other two scenarios, which leads to an adverse effect on training extra networks in CDS”. I can understand the former (“the number of agents in this scenario is much higher than that in the other two scenarios”), but the latter (“which leads to an adverse effect on training extra networks in CDS”) was unclear. What did the authors try to mention?
* Figure 1: How did the authors create the Skill_ID pool? I cannot find this. In particular, I want to know whether this is created in a rule-based or data-driven manner.
* L142- “This procedure is essentially a many-to-many assignment problem”. However, I cannot find how to solve this problem. How did the authors solve this problem?
* Appendix A: Eq. (12) is different from eq. (4). If s can be replaced with o, but what is x? Where is the 3rd term of eq. (4)?
In Appendix Table 2, n_skill of Hard Counter-attack in GRF was 3, but in Appendix C.1 Skill Demonstration on GRF, the authors trained HSL with 4 skills. Which is correct?


**Limitations:**

The limitations and potential negative societal impact were described in Appendices D and E, respectively. However, the latter was not concrete (“Therefore, we do not anticipate a direct negative outcome. In practical applications involving our method, potentially negative outcomes might occur.”).

**Strengths And Weaknesses:**

The strength of the paper is as follows:
* Novelty: The authors proposed the skill representation mechanism with an auto-encoder model, the skill selector to realize the assignment from agents to skills, and the skill-based policy learning mechanism with a mutual information-based intrinsic reward function.
* Clear results: Experimental results show that the framework obtains the best performance on three challenging benchmarks, i.e., StarCraft II micromanagement tasks, Google Research Football and GoBigger, over state-of-the-art MARL methods.

The weakness of the paper is as follows (please also see below “Questions):
* The presentation was sometimes inconsistent.
* The method descriptions were sometimes unclear.

---

> ### Author Response · Authors · 2022-08-02
> **Response for Reviewer amao (Part I)**
>
> Thank you very much for your detailed and constructive comments. Below please find the responses to some specific comments.
>
> **Q1:** L45- “However, extra introduced networks for all agents hinder the application of CDS on large-scale tasks, which will be illustrated in the experiment.” The proposed HSL requires extra networks to obtain the skill diversity. Since HSL outperformed CDS in MARL performance, the problem of CDS can be mentioned differently
>
> **A1:** Our approach does require extra network which is responsible for skill discovery. But the extra network in CDS is not the same as the extra network in our method. The extra network in CDS refers to the policy learning network. Previous MARL methods introduce the parameter sharing mechanism to reduce the policy search space. The parameter sharing mechanism uses only one policy network for all agents. However, this causes the problem of learning similar policies for different agents. In order to alleviate this problem, CDS adds an extra policy learning network for each agent. Note that the parameter sharing mechanism is not applied to extra networks. The final policy of each agent is the weighted sum of the outputs of agent’s two policy networks. For large-scale tasks, the number of extra networks added by CDS increases linearly with the number of agents, which will greatly degrade the performance of CDS. In the method HSL, we apply the parameter sharing mechanism both in the skill selector and the skill policy learning model. The number of neural networks will not increase with the number of agents, which is the reason that HSL performs better than CDS. Besides, we will include some of these details and make this part clearer in the revised version.
>
> ---
>
> **Q2:** L248- In association with the above, “Still, the reason is that the number of agents in this scenario is much higher than that in the other two scenarios, which leads to an adverse effect on training extra networks in CDS”. I can understand the former (“the number of agents in this scenario is much higher than that in the other two scenarios”), but the latter (“which leads to an adverse effect on training extra networks in CDS”) was unclear. What did the authors try to mention?
>
> **A2:** As mentioned in A1, the parameter sharing mechanism is often used in MARL methods. In order to address the problem of learning similar policy for agents in the parameter sharing mechanism, CDS adds an extra policy network for each agent to learn heterogeneous policies. The policy network of each agent in CDS is consists of the shared policy network and the individual policy network. The number of individual policy networks need to be trained equals to that of agents in multi-agent tasks. Suppose a large-scale task contains $N$ agents, the typical MARL method QMIX trains only one shared policy network while CDS needs to train $N+1$ policy network. Obviously, the training efficiency of CDS is much lower than that of QMIX, which is called an adverse effect on training extra networks in CDS.
>
> ---
>
> **Q3:** Figure 1: How did the authors create the Skill_ID pool? I cannot find this. In particular, I want to know whether this is created in a rule-based or data-driven manner
>
> **A3:** The Skill_ID pool contains one-hot vectors of skill-ids. The creation of the Skill_ID pool is simple. For example, if a total of 4 skills are learned, we first set up the skill-id list $\\{1,2,3,4\\}$. Then we encode each element in the skill-id list into one-hot vectors and we get $\\{0001,0010,0100,1000\\}$. Finally, we add these one-hot vectors into the Skill_ID pool. One of the features of our HSL is the automatic skill learning. Therefore, we cannot use prior information and expert knowledge to construct skill features or create the Skill_ID pool. We just set up simple one-hot vectors of skill-ids and use these vectors as skill features to learn skill policies automatically. The more these skill features differ from each other, the better. Therefore, the orthogonal one-hot vectors of skill-ids is a very appropriate choice. The creation of the Skill_ID pool is a simple rule-based manner, and the rule is to ensure orthogonality between features in the Skill_ID pool as much as possible.
>
> ---
>
> **Q4:** L142- “This procedure is essentially a many-to-many assignment problem”. However, I cannot find how to solve this problem. How did the authors solve this problem?
>
> **A4:** Thanks very much for your constructive comments. We think that we are not using the accurate words. The assignment problem you are referring to should be a particular case of transportation problem where the objective is to assign a number of resources to an equal number of activities so as to minimize total cost or maximize total profit of allocation. The problem mentioned in L142 is that skill selector chooses proper skills for different agent at the same time. However, the statement in L142 is not strict. We will use more appropriate word in the revised version.

---

> > ### Author Response · Authors · 2022-08-02
> > **Response for Reviewer amao (Part II)**
> >
> > **Q5:** Appendix A: Eq. (12) is different from eq. (4). If s can be replaced with $o$, but what is $x$? Where is the 3rd term of eq. (4)? In Appendix Table 2, $n_{skill}$ of Hard Counter-attack in GRF was 3, but in Appendix C.1 Skill Demonstration on GRF, the authors trained HSL with 4 skills. Which is correct?
> >
> > **A5:** Thanks very much for your constructive comments. Equation (12) in Appendix A is not consist with Equation (4) in the main body because Equation (12) is a previous wrong version. Equation (12) in Appendix A is a more general case. We replace partially observation features o in Equation (4) with state features s. But x in the second element of the first term is wrong. The correct one is state features $s$. Missing the third term in Equation (4) is also wrong. The correct Equation (12) includes the third term. The correct value of $n_{skill}$ of Hard Counter-attack in GRF is 4. This scenario is harder than the *3_vs_1_with_keeper* scenario. Therefore, Hard Counter-attack   scenario requires more skills than *3_vs_1_with_keeper* scenario does. We will correct these issues in the revised version.
> >
> > ---
> >
> > **Q6:** The limitations and potential negative societal impact were described in Appendices D and E, respectively. However, the latter was not concrete (“Therefore, we do not anticipate a direct negative outcome. In practical applications involving our method, potentially negative outcomes might occur.”).
> >
> > **A6:** Here we add some examples to show potentially negative outcomes of our method. The first is related to human oversight of the MARL system. The Ethics Guidelines for Trustworthy AI report published in 2019 by the European Commission’s High Level Expert Group on AI states AI systems have to allow for human oversight in order to support human autonomy and decision-making.  However, the data flow in a DRL system is acting upon may be incomprehensible to humans, or simply too large and fast-moving for meaningful oversight to be maintained. As for MARL methods, understanding and intervening at the level of a single agents is a more difficult problem, which poses additional challenges for oversight. The other is MARL agents apply trial-and-error approach to explore the environment to discover actions that lead to the highest reward over time. However, this is unacceptable in many real-world contexts. For instance, we cannot have self-driving cars running over pedestrians, or an energy control system accidentally switching off electricity in a hospital, before learning not to do these things.

---

> > > ### Comment · Reviewer_amao · 2022-08-09
> > > **The reviewer's response**
> > >
> > > Thank you for the response.
> > > Totally, unclear points for me are clarified.

---

### Official Review · Reviewer_j8Gw · 2022-07-11

**Rating:** 4
**Confidence:** 4
**Soundness:** 3 good
**Presentation:** 2 fair
**Contribution:** 3 good

**Summary:**

The authors suggest a skill-based multi-agent learning algorithm that generates diverse skills for finding heterogeneous behaviours. In the proposed mehtod, there are three mechanisms for capturing heterogeneous skills: skill representation, skil selector and skill-based policy learning. Experiment results show that the proposed method outperformed than other multi-agent methods on three multi-agent RL benchmarks.


**Questions:**

   1. Paragraphs on lines 58-70 should go to the relevant work section.

   2. There is no description how to compute the local $Q_i^v$ in Secion 3.2

   3. The loss function for $q_{\theta_z}$ is missing.

   4. As previously explained, the reviewer believes that the skill-id is sufficient to be used as a latent skill variable. Could you provide the results of using skill-ids as skill varaibles instead of skill representation method.

**Limitations:**

yes

**Strengths And Weaknesses:**

Strenghts

  1. The proposed method (HSL) suggests two types of intrinsic rewards for skill selector and policy learning to enable effective exploration on both skill selection and heterogeneous behaviour.

  2. Experiment results show the significant improvements on several benchmarks.

Weakenesses
  1. The reviewer does not understand why the skill representation mechanism is needed in the proposed framework.
     1) Two decoders can be trained without the latent skill varialbe, since two decodes already take observation and actions. So I could not agree that the auto-encoder generates latent skill variables that reveal different effect of different skills.
     2) In equation (2), the cosine distance is used to distinguish latent skill varialbes, but I think the skill-id is already enough for distinguishability.

  2. The motivation of introducing $H[a|o,z]$ for intrinsic reward $r_m$ is not acceptable. The reviewer thinks that the effect of maximizing policy entorpy is just effective exploration not discrimability of skill.

  3. There is something wrong with constructing the lower bound of the intrinsic reward $r_m$.

        In equation 4, two terms 1 and 2 are conditional entropy, but the author use these conditional entropy as just common shannon entropy.
        1) $H[a|z, o] =  \mathbb{E}_{p(z|o)}[H(p(a|z,o))] \neq H(p(a|z,o))$ is the conditional entropy between random variable $a$ and $z$ given $o$.
        2) $H[z|o] =\mathbb{E}_{p(o)}[H(p(z|o))] \neq H(p(z|o)) $  is the conditional entropy between random variable $z$ and $o$

---

> ### Author Response · Authors · 2022-08-02
> **Response for Reviewer j8Gw (Part I)**
>
> Thank you for the detailed and constructive comments. Below please find the responses to some specific comments.
>
> **Q1:** The reviewer does not understand why the skill representation mechanism is needed in the proposed framework
>
> **A1:** The skill representation mechanism essentially generates skill latent variable $z_{latent}$ which implies certain correlations among latent skill variables, reward function and state transition function through the training of the proposed auto-encoder structure. $z_{latent}$ is important for the distinguishable of agents for skill selection, which plays an indirect role in improving the discriminability for the skill learning. For the skill selector, $z_{latent}$ and the observation latent are both used as the input feature. Due to the implied correlation, $z_{latent}$could greatly improve the learning efficiency of the skill selector. In fact, we find that the skill selector works not well with only the observation latent.
>
> ---
>
> **Q2:** Two decoders can be trained without the latent skill variable, since two decodes already take observation and actions. So I could not agree that the auto-encoder generates latent skill variables that reveal different effect of different skills
>
> **A2:** The reason of combining latent skill variables with observations and actions is that embedding information from the environment into latent skill variables. The encoder in the auto-encoder model transforms the one-hot variables of skill-ids into latent skill variables. However, only with latent skill variables extracted from the encoder, the skill selector cannot assign proper skill for agents because it does not know how latent skill variables related to rewards and the state transition procedure of the environment. In the implementation of the auto-encoder, we concatenate observations, actions and latent skill variables extracted from the encoder, which is then used as the input feature for the two decoders. We train the decoders with supervised reward and next observations from the environment, which embeds implicated relations among latent skill variables, reward function and state transition function into $z_{latent}$.
>
> ---
>
> **Q3:** In equation (2), the cosine distance is used to distinguish latent skill variables, but I think the skill-id is already enough for distinguishability
>
> **A3:** As mentioned in A1 and A2, the skill selector requires not only the observation latent feature but also the skill latent feature. The skill latent feature can be replaced with the skill-id. The different skill-ids are orthogonal to each other, which ensures the distinguishability of these features. However, compared to the latent skill variables, the skill-id lacks the encoding of environmental information. This leads to the fact that the skill selector can only obtain the agents’ perception of the environment and cannot know the interaction between the skill policy and the environment. Therefore, the latent skill variables are very important for the skill selector and cannot be simply replaced by the skill-id. In our implementation, we found that if the cosine distance is not constrained, latent skill variables will tend to be similar with a certain probability during the training process. Thus, we add the constraint of the cosine distance in Equation (2) in order to ensure the distinguishability of latent skill variables.
>
> ---
>
> **Q4:** The motivation of introducing $H[a|o,z]$ for intrinsic reward rm is not acceptable. The reviewer thinks that the effect of maximizing policy entorpy is just effective exploration not discrimability of skill
>
> **A4:** Thank the reviewer for pointing out this problem. In fact, the word “discrimability” is not used exactly in this paper. Diversity is more proper here. Our aim of introducing $H[a|o,z]$ is to encourage the diversity of skills, which incentivizes the skills to be as diverse as possible by learning skills that act as randomly as possible. This is consistent with the aim of effective exploration.
>
> ---
>
> **Q5:** There is something wrong with constructing the lower bound of the intrinsic reward rm. In equation 4, two terms 1 and 2 are conditional entropy, but the author use these conditional entropy as just common shannon entropy.
>
> **A5:** Thank the reviewer for the detailed check. In fact, the lower bound for the intrinsic reward is correct. Some formulas are not precisely written, and here we give more explanations.
>
> Term 1 and term 2 in Equation (4) are conditional entropy and we cannot simply eliminate these terms. After careful formula derivation, the correct method for constructing the lower bound of the intrinsic reward can be described as follows:
>
> \begin{aligned}
>     r^m=&I(z;o)+I(a;z|o)+H[a|z,o] \\\\
>     =&(H[z]-H[z|o])+(H[a|o]-H[a|z,o])+H[a|z,o] \\\\
>     =&H[z]-H[z|o]+H[a|o] \\\\
>     \geq&H[p(z)]-\mathbb{E}_\{p(a|o)\}[H[p(z|a,o)]]+\mathbb{E}_\{p(z|o)\}[H[p(a|z,o)]]-H[z|o]+  \mathbb E_\{p(a,z|o)\}[\log \frac{1}{p(z|o)}]
> \end{aligned}

---

> > ### Author Response · Authors · 2022-08-02
> > **Response for Reviewer j8Gw (Part II)**
> >
> > Now we talk about how to eliminate term $ -H[z|o]+\mathbb{E}_\{p(a,z|o)\} [\log \frac{1}{p(z|o)}]$.
> >
> > For $H[z|o]$, we get:
> > \begin{aligned}
> > H[z|o]=&\sum_\{o'\} p(o')H[z|o=o']\\\\
> > =&\sum_\{o'\}p(o')\sum_\{z'\} p(z'|o') \log \frac{1}{p(z'|o')}\\\\
> > =&\sum_\{o'\}\sum_\{z'\}p(o',z')\log \frac{1}{p(z'|o')}\\\\
> > =&\sum_\{a'\}p(a')\sum_\{o'\}\sum_\{z'\}p(o',z')\log \frac{1}{p(z'|o')}\\\\
> > =&\sum_\{a'\}\sum_\{o'\}\sum_\{z'\}p(a',o',z')\log \frac{1}{p(z'|o')}
> > \end{aligned}
> >
> > For $\mathbb{E}_\{p(a,z|o)\} [\log \frac{1}{p(z|o)}]$, we get:
> > \begin{aligned}
> > \mathbb{E}_\{p(a,z|o)\} [\log \frac{1}{p(z|o)}]=&&\sum_\{a'\}\sum_\{o'\}\sum_\{z'\} \frac{p(a',o',z')}{p(o')}\log \frac{1}{p(z'|o')}
> > \end{aligned}
> >
> > Then we can get:
> > \begin{aligned}
> > 0\leq p(o') \leq 1 \Rightarrow & \frac{1}{p(o')} \geq 1 \Rightarrow \frac{p(a',o',z')}{p(o')} \geq p(a',o',z') \Rightarrow \mathbb{E}_\{p(a,z|o)\} [\log \frac{1}{p(z|o)}] \geq H[z|o]\\\\
> > \Rightarrow & -H[z|o]+\mathbb{E}_\{p(a,z|o)\} [\log \frac{1}{p(z|o)}] \geq 0
> > \end{aligned}
> >
> > In summary, we can get $r^m \geq H[p(z)]-\mathbb{E}_\{p(a|o)\}[H[p(z|a,o)]]]+\mathbb{E}_\{p(z|o)\}[H[p(a|z,o)]]$, which is the lower bound in our paper. We will fix the constructing of the lower bound of the intrinsic reward in the revised version.
> >
> > ---
> >
> > **Q6:** Paragraphs on lines 58-70 should go to the relevant work section
> >
> > **A6:** This paragraph forms a connecting link between the preceding and the following. The previous paragraph introducing the skill learning problem, and then this part introduces some similar works. It explains how those works deal with the skill learning problem and points out their limitations. The possible room for improvement is also described, which provides preparation and groundwork for the method proposed in the following paragraphs.
> >
> > However, this paragraph involves many details of previous work, we will organize these details and put them into the related work section in the revised version.
> >
> > ---
> >
> > **Q7:** There is no description how to compute the local $Q_i^v$ in Secion 3.2
> >
> > **A7:** The computation of the local $Q_i^v$ is shown in Figure 1 but omitted in Section 3.2. We explain it in detail here. As shown in Figure 1, the computation of Qiv is simple. Once we have latent observation variables of all agents and latent skill variables of all skills, we represent these two variables as matrices. The dimensions of matrices transformed from latent observation variables and latent skill variables are *(num_agents, num_latent_dim)* and *(num_skills, num_latent_dim)*, respectively. Then we perform transpose operation on the second matrix. Finally, we apply matrix multiplication of these two matrices and get the result $Q^v$ with the dimension of *(num_agents, num_skills)*. Row $i$ in $Q^v$ is the local $Q_i^v$ of agent $i$ and each element $j$ in $Q_i^v$ means the probability of choosing skill $j$.
> >
> > ---
> >
> > **Q8:** The loss function for $q_{θ_z}$ is missing
> >
> > **A8:** In the implementation, we adopt variational inference to estimate $p(z|o,a)$ and learn the variational distribution $q_{θ_z}(z|o,a)$. The variational inference method is proposed in Agakov, 2004, which is not the main contribution of our paper. We could add these details into the appendix. $q_{θ_z}(z|o,a)$ is parameterized by a neural network and $θ_z$ are parameters of this neural network. This network takes observations and actions of agents as inputs and outputs the estimation of chosen skill $z$. We apply supervised training of this network, with the actual skill selected as the supervision. Therefore, the loss function of $q_{θ_z}$ is $MSE(z_\{out\}, z_\{true\})$, where MSE represents the MSE loss, $z_\{out\}$ is the output of the network and $z_\{true\}$ denotes the actual chosen skill $z$.
> >
> > ---
> >
> > **Q9:** As previously explained, the reviewer believes that the skill-id is sufficient to be used as a latent skill variable. Could you provide the results of using skill-ids as skill varaibles instead of skill representation method
> >
> > **A9:** We have conducted experiments of replacing latent skill variables. Results are shown in the following table.
> > |  Scenario   | 5m_vs_6m  | MMM2  | 27m_vs_30m  | Corridor  | 3s5z_vs_3s6z  | 6h_vs_8z  |
> > |  ----  | ----  | ----  | ----  | ----  | ----  | ----  |
> > | HSL no skill repr  | 51% |  91% | 84% | 35% | 62% | 44% |
> > | HSL skill-id  | 67% | 92% | 93% | 54% | 73% | 61% |
> > | HSL  | 85% | 100% | 100% | 82% | 84% | 82% |
> >
> > |  Scenario   | 3_vs_1_with_keeper  | Hard Counter-attack | Corner  | GoBigger:3_vs_3  | GoBigger:3_vs_3 _with_thorn  |
> > |  ----  | ----  | ----  | ----  | ----  | ----  |
> > | HSL no skill repr  | 42% |  28% | 17% | 0% | 0% |
> > | HSL skill-id  | 55% | 37% | 33% | 14% | 26% |
> > | HSL  | 70% | 65% | 56% | 38% | 55% |

---

> > > ### Author Response · Authors · 2022-08-02
> > > **Response for Reviewer j8Gw (Part III)**
> > >
> > > These two tables show the extended ablation experiment of the skill representation mechanism where HSL no skill repr is the same in ablation study in the main body and HSL skill-id replaces the latent skill variables in the skill selector with skill-ids. We can observe that HSL with latent skill variables in the skill selector achieves the best performance among all scenarios. This shows that latent skill variables play an important role in the skill selector because these variables contain information of the environmental model mentioned in A1. HSL skill-id outperforms HSL no skill repr because skill-ids are orthogonal to each other. Orthogonality ensures the distinguishability of the skill-id, which is useful for skill selection. HSL no skill repr uses raw states in the skill selector and does not includes any distinguishable features for skills. Therefore, it gets the lowest win rates. The lack of information about the environment model makes the quality of skill selection in HSL HSL skill-id is lower than that of HSL skill-id. This is the reason for the performance gap between these two methods.
> > >
> > > [1] David Barber Felix Agakov. The im algorithm: a variational approach to information maximization. Advances in Neural Information Processing Systems, 16:201, 2004.

---

> > > ### Comment · Reviewer_j8Gw · 2022-08-08
> > > **Response to authors**
> > >
> > > Thanks for answers to my feedback. The most of concerns are solved, but the derivation of Intrinsic reward isn't still clear.
> > >
> > > 1. The value $\mathbb{E}_{p(a,z|o)}[\log \frac{1}{p(z|o)}]$ is expectation over the variables $a,z$ not $o$.
> > >
> > >      \begin{align}
> > >           \sum_{a'} \sum_{z'} \frac{p(a',o,z')}{p(o)}\log \frac{1}{p(z'|o)}
> > >      \end{align}
> > >
> > > 2.  When the authors find the lower bound on $H[a|s]$ in Theorem 1(Appendix A), the authors should take expectation over the $s$. Then the lower bound will become
> > >
> > >      \begin{align}
> > >           \mathbb{E}_{p(a,z,s)}\Big[\log\frac{p(z|a,s)}{p(a,z|s)}\Big]
> > >      \end{align}

---

> > > > ### Author Response · Authors · 2022-08-08
> > > > **Response to the further questions**
> > > >
> > > > Thanks for your positive feedback on our work and for your valuable comments.
> > > >
> > > > Inspired by your second comment and the derivation of $H[z|o]$ in our last response, we find an easier way to construct the lower bound of the intrinsic reward. In this way, we do not have to generate the term $\mathbb E_{p(a,z|o)}[\log \frac{1}{p(z|o)}]$ mentioned in your first comment. Further, we make a careful check on the derivation of the lower bound of the intrinsic reward. Your concerns are answered below.
> > > >
> > > > Firstly, we can construct the correct lower bound of $H[a|s]$ according to the second comment.
> > > > \begin{align}
> > > > H[a|s] =& \sum_{s} \sum_{a} p(a,s) \log \frac{1}{p(a|s)} \\\\
> > > > \geq& \mathbb{E}_\{p(a,z,s)\}\Big[\log\frac{p(z|a,s)}{p(a,z|s)}\Big] \\\\
> > > > =& \mathbb{E}_\{p(a,z,s)\}\Big[\log p(z|a,s)\Big] + \mathbb{E}_\{p(a,z,s)\}\Big[\frac{1}{\log p(a,z|s)}\Big]  \\\\
> > > > =& \mathbb{E}_\{p(a,s)\}\Big[\mathbb{E}_\{p(z|a,s)\} \log p(z|a,s)\Big] + \mathbb{E}_\{p(a,z,s)\}\Big[\frac{1}{\log p(z|s)p(a|z,s)}\Big] \\\\
> > > > =& -\mathbb{E}_\{p(a,s)\}\Big[H[p(z|a,s)]\Big] + \mathbb{E}_\{p(a,z,s)\}\Big[\frac{1}{\log p(z|s)}\Big] + \mathbb{E}_\{p(a,z,s)\}\Big[\frac{1}{p(a|z,s)}\Big] \\\\
> > > > =& -\mathbb{E}_\{p(a,s)\}\Big[H[p(z|a,s)]\Big] + \mathbb{E}_\{p(a,z,s)\}\Big[\frac{1}{\log p(z|s)}\Big] + \mathbb{E}_\{p(z,s)\}\Big[H[p(a|z,s)]\Big]
> > > > \end{align}
> > > >
> > > > Then we detail the derivation of the lower bound of the intrinsic reward:
> > > > \begin{align}
> > > > r^m=&I(z;o)+I(a;z|o)+H[a|z,o] \\\\
> > > > =&(H[z]-H[z|o])+(H[a|o]-H[a|z,o])+H[a|z,o] \\\\
> > > > =&H[z]-H[z|o]+H[a|o] \\\\
> > > > \geq&H[p(z)]-\mathbb{E}_{p(a,o)}\Big[H[p(z|a,o)]\Big]+\mathbb{E}_\{p(z,o)\}\Big[H[p(a|z,o)]\Big] -H[z|o]+ \mathbb E_\{p(a,z,o)}\Big[\log \frac{1}{p(z|o)}\Big]
> > > > \end{align}
> > > >
> > > > Now we focus on the last term $-H[z|o]+ \mathbb E_\{p(a,z,o)}\Big[\log \frac{1}{p(z|o)}\Big]$.
> > > > \begin{align}
> > > > \mathbb E_\{p(a,z,o)}\Big[\log \frac{1}{p(z|o)}\Big] =& \sum_{z} \sum_{o} \sum_{a} p(z,o, a) \log \frac{1}{p(z|o)} \\\\
> > > > =& \sum_{z} \sum_{o} \sum_{a} p(a)p(z,o | a) \log \frac{1}{p(z|o)} \\\\
> > > > =& \sum_{z} \sum_{o} p(z,o) \log \frac{1}{p(z|o)} \\\\
> > > > =& H[z|o]
> > > > \end{align}
> > > >
> > > > Therefore, we can eliminate the term $-H[z|o]+ \mathbb E_\{p(a,z,o)}\Big[\log \frac{1}{p(z|o)}\Big]$. In summary, we can get the lower bound of the intrinsic reward in our paper in this easier way.  We will fix the constructing of the lower bound of the intrinsic reward in the revised version.
> > > >
> > > > Thanks again for your inspiration and great suggestions!

---

> > > > ### Author Response · Authors · 2022-08-10
> > > > **Supplementary notes for Reviewer j8Gw**
> > > >
> > > > Thank you very much for the inspiring and insightful comments which really helped us to improve our work!
> > > > We have submitted a revised paper. We hope that we have addressed all your concerns.
> > > > Please let me know if you have any other issues.

---

### Official Review · Reviewer_6BRg · 2022-07-19

**Rating:** 6
**Confidence:** 4
**Soundness:** 2 fair
**Presentation:** 3 good
**Contribution:** 3 good

**Summary:**

The paper proposes a framework for multi-agent reinforcement learning with the goal of selecting heterogeneous behaviors and allocating them to agents so that an optimal policy can be achieved. The proposed framework represents skills as latent variables which are used to assign skills so that in the end agents can learn heterogeneous policies. The proposed framework is tested on three different multi-agent tasks and compared with other methods. In addition, an ablation study is included.

**Questions:**

In addition to commenting to the two points raised in "Strengths and weaknesses" -- i.e., the non direct applicability to complex scenarios and the comparison with non-MARL classic methods:

A few details that could make the paper clearer:
- the methods compared appear to be from the same group; it is worth to include and/or discuss the choice for the comparison. For example, MAVEN could be used as comparison.
- the differences between the scenarios in each multi-agent task are not clear, as only some arbitrary naming is used.
- it is worth to make examples of skills and actions as the way the paper discusses them they appear to be the same.

There are a few language problems, including:
- "attracts widely attention" -> "attracts wide attention"
- space after ; -- e.g., "are represented;(2)" -> "are represented; (2)"
- "dispose of" -> "consider"
- "With the representation" -> "With this representation"
- "one-hot" -> "one-shot"
- "should explore differently and access different states of the environment." There is different repeated, and "explore differently" could be more specific.
- "described as follows Equation (10)." -> "described as follows."
- "comparing results" -> "comparative results"
- "Another notification" -> "Another note"
- "sparce" -> "sparse"

**Limitations:**

The appendix includes a section where the main limitations are discussed and providing some some potential venues of future work. It is worth to discuss how the framework could scale to more complex real-world scenarios.

The section about the impact discusses that overall the method itself should not have a direct negative outcome, however notes that in practical applications they could occur. Some examples and ways to address them could be discussed.

**Strengths And Weaknesses:**

The paper appears to inspiration from a few other papers that are cited, including [4] and [32], but the view taken by the paper in identifying skills to improve MARL is significant to potentially achieve cooperative tasks. The paper is also overall well written, providing the intuition on the main choices made for the proposed framework and presenting sound technical details. It is appreciated also the experiments in three different problems and the evaluation with other methods, as well as the ablation study.

While the paper shows an interesting direction, the tasks are relatively constrained making the significance of the proposed framework more limited. Indeed, as noted also in the limitations, the framework does not appear to be able to learn appropriate policies when more skills are present.

In addition, the paper could discuss not only the literature on MARL, but also classic methods of heterogeneous multi-agent task allocation, which is currently missing, including
- Wu, H., Ghadami, A., Bayrak, A. E., Smereka, J. M., & Epureanu, B. I. (2021). Impact of Heterogeneity and Risk Aversion on Task Allocation in Multi-Agent Teams. IEEE Robotics and Automation Letters, 6(4), 7065-7072.
- Korsah, G. A., Stentz, A., & Dias, M. B. (2013). A comprehensive taxonomy for multi-robot task allocation. The International Journal of Robotics Research, 32(12), 1495-1512.
- Emam, Y., Mayya, S., Notomista, G., Bohannon, A., & Egerstedt, M. (2020, May). Adaptive task allocation for heterogeneous multi-robot teams with evolving and unknown robot capabilities. In 2020 IEEE International Conference on Robotics and Automation (ICRA) (pp. 7719-7725). IEEE.
- Schillinger, P., Bürger, M., & Dimarogonas, D. V. (2018). Simultaneous task allocation and planning for temporal logic goals in heterogeneous multi-robot systems. The international journal of robotics research, 37(7), 818-838.

---

> ### Author Response · Authors · 2022-08-02
> **Response for Reviewer 6BRg (Part I)**
>
> We sincerely thank you for your time and efforts. Below please find the responses to some specific comments.
>
> **Q1:** While the paper shows an interesting direction, the tasks are relatively constrained making the significance of the proposed framework more limited.
>
> **A1:**  The tasks in this paper are representative benchmarks of partially observable multi-agent games in MARL methods, for example, in QMIX, RODE, CDS etc. Most work in this research area takes those tasks to conduct experiments. In fact, those tasks are not easy to be tackled. Below we give some descriptions on those tasks.
>
> All scenarios in SMAC environment are gradually conquered by MARL algorithm in recent years. Therefore, we introduce more complex environments which are GRF and GoBigger. The difficulty of both these two environments lies in the randomness of the opponent's policy which is built-in strong rules. The strong randomness policies pose a huge challenge for MARL algorithms to learn effective policies. Our algorithm achieves STOA performance on all three environments, which indicates that our algorithm has made great progress compared to other MARL algorithms in this field.
>
> Another very popular and skill-related MARL research field is multi robot control. However, the action spaces of the agents in the robot environment are continuous, while those in the problem we are working on are discrete action spaces. Our work tries to learn skill policies on discrete action spaces, which is different from the robot control environment.
>
> ---
>
> **Q2:**  Indeed, as noted also in the limitations, the framework does not appear to be able to learn appropriate policies when more skills are present.
>
> **A2:** As long as the number of skills does not exceed a threshold, our proposed framework can still learn appropriate policies when the number of skills increases. This is proven in experimental results in Figure 5 in Appendix. Our framework is organized as a bi-level learning structure. The policy searching space becomes huge as the number of skills increases. All MARL methods will counter such problem when the task is too complex. Further, increasing the scalability of a MARL method on more complex tasks is always one of the common research directions. It would also a good direction to extend our method for other more complex tasks.
>
> ---
>
> **Q3:** In addition, the paper could discuss not only the literature on MARL, but also classic methods of heterogeneous multi-agent task allocation
>
> **A3:** Typical multi-agent task allocation assigns the decomposed subtasks to the agents and arranges the execution order of the subtasks properly according to the priority or constraint relationship of the subtasks. Typical heterogeneous multi-agent task allocation must first split the task into subtasks and then pre-define the priority and the costs of performing tasks and communication. Different from the typical task allocation problem, multi-agent tasks in our paper can be modeled as a partially observable Markov decision process (POMDP). There is no priority and the costs of performing tasks and communication in POMDP. Therefore, we design skill learning method to automatically discover skills without any pre-definition. In the skill selector, the only information we know is the latent skill variables extracted from the skill representation mechanism. It is obviously different from the typical task allocation problem. Therefore, we cannot apply classic methods of heterogeneous multi-agent task allocation on the skill selection procedure.
>
> ---
>
> **Q4:**  the methods compared appear to be from the same group; it is worth to include and/or discuss the choice for the comparison. For example, MAVEN could be used as comparison.
>
> **A4:** The methods compared in experiments are selected from different aspects of the MARL research. The first group includes value decomposition methods such as QMIX and QPlex. These methods designs a suitable model structure and efficient training methods. The second group contains role-based methods such as ROMA and RODE. The idea of these methods is that a MA task can be divided into several sub-tasks. They introduce the concept of role to tackle these sub-tasks, which improves the efficiency of solving the total task. Methods in the third group aim to learn diversity policies. CDS designs a diversity-based approach and HSD introduces a simple skill mechanism. MAVEN, belonging to the third group, encourages effective exploration by learning a hierarchical policy to condition agents’ policies. Our proposed method belongs to the third group as well. Our method designs effective skill discovery mechanism and skill policy learning mechanism and outperforms HSD.

---

> > ### Author Response · Authors · 2022-08-02
> > **Response for Reviewer 6BRg (Part II)**
> >
> > **Q5:** the differences between the scenarios in each multi-agent task are not clear, as only some arbitrary naming is used.
> >
> > **A5:** Here we give more explanations for the differences on all used scenarios.
> >
> > We choose six maps in SMAC, i.e., *MMM2, 3s5z_vs_3s6z, 5m_vs_6m, 27m_vs_30m, corridor* and *6h_vs_8z*. The first two maps are heterogeneous maps with different agents. The rest four maps are homogeneous maps with same agents. It is important to learn heterogeneous policies to beat enemies in these maps.
> >
> > For the GRF environment, we select three scenarios which are *3_vs_1_with_keeper, Hard_counter_attack* and *Corner*. Agents in these scenarios are homogeneous, resulting in similar observations and action spaces. The scenario *3_vs_1_with_keeper* contains three players and *Hard_counter_attack* is a more difficult scenario with four players to be controlled. *Corner* scenario contains all players in a football match.
> >
> > In GoBigger environment, we choose *3_vs_3* and *3_vs_3_with_thorn* for experiments. These two scenarios are very similar, except that the latter has four thorn balls. The existence of thorn balls brings greater uncertainty to the scenario and increases the difficulty for policy learning in MARL methods.
> >
> > ---
> >
> > **Q6:** it is worth to make examples of skills and actions as the way the paper discusses them they appear to be the same.
> >
> > **A6:** The skill is the conditioned policy $\pi(A|S,Z)$. The action generated from the skill is based on both the agent’s observation and its selected skill feature. The action generated from the vanilla policy is only based on the agent’s observation. The skill is more like a mask on original action space when generating skill policies. For example, agents can run in four directions and attack one of enemies in SMAC. The running skill works like a mask, the four directions of running are optional in this skill, while attacking one of the enemies is not optional. For the attacking skill, attacking one of the enemies is optional while the four directions of running are not optional. If the appropriate skill assignment can be found, we can achieve efficient policy learning because the policy searching space is reduced. Moreover, another benefit of the skill is the ease of learning heterogeneous policies.
> >
> > ---
> >
> > **Q7:**  It is worth to discuss how the framework could scale to more complex real-world scenarios
> >
> > **A7:** We consider the scale our framework to a real-world multi-agent autonomous driving task. In this task, each agent needs to not only drive the vehicle, but also interact with vehicles controlled by other agents. Interact between vehicles includes changing lanes, merging and overtaking other vehicles. Therefore, skills can be clearly defined according to the kind of interaction between the agents. Initial training is performed on a simulator, which has many rules and available data. Therefore, to speed up the training process and improve robustness, imitation learning can be used to quickly learn skill policies for different scenarios. When the training on the simulator converges, the learned policies learned need to be migrated to the real-world environment using the sim2real approach. Methods such as domain transformation and transfer learning can be used. Finally, the completed training algorithm framework along with the rule system is deployed to the real-world environment and the fine-tuning technique of our framework is performed on a real-world multi-agent autonomous driving task to finally achieve the desired results.
> >
> > ---
> >
> > **Q8:** The section about the impact discusses that overall the method itself should not have a direct negative outcome, however notes that in practical applications they could occur. Some examples and ways to address them could be discussed
> >
> > **A8:** Here we give two examples to show the potential negative outcome of our method in practical applications.
> >
> > The first is related to the human oversight of a MARL system. Trusty AI systems must allow for human oversight to support human autonomy and decision-making. However, MARL applications may pose challenges to human oversight because these applications aim to increase the autonomy of machines. For example, a MARL system for monitoring and adjusting energy usage in a building may be constantly making so many small decisions. These decisions are difficult for a human to review and change decisions after the fact. One way to address this problem is to impose constraints while the system is being designed such as waning a human if levels go beyond a certain threshold.
> >
> > The second is related to the security of a MARL system. For example, even a demonstrably safe MARL-based multi robot system could be forced into dangerous collision scenarios by perturbing its sensory input or disrupting its reward function. Possible ways to address is to oversee the transparency of the training data and the reward function or to develop safe multi-agent reinforcement learning methods.

---

### Meta-Review · Area_Chair_pqqX · 2022-08-30

**Recommendation:** Accept
**Confidence:** Less certain

**Metareview:**

The reviewers have largely agreed upon the value of the paper's concept (heterogeneous skills for MARL) and appreciated its impressive experimental gains on a range of environments. Each reviewer pointed out unique areas for improvement: citations to classical work, precision in derivations around conditional entropy and general writing improvements - which I find were largely addressed in the rebuttal discussions.

Although the precise notion of skills and their application in the MA setting can be debated (is it just an exploration guide, as stated by reviewer j8Gw?) that is quite an interesting debate to engage in. The relatively unique concept for the area and strong performance on a core set of benchmarks will be quite interesting for the NeurIPS community.

**Award:**

No

---

### Decision · Program_Chairs · 2022-09-14

Accept